# Covalent modification of a glutamic acid inspired by HaloTag technology

Ruirui Zhang [1,6], Jie Liu [1,6], Raphael Gasper [2], Anke Unger [3], Farnusch Kaschani [4], Markus Kaiser [4], Petra Janning[1] & Herbert Waldmann [1,5] ✉

For targeted covalent protein modification at low-reactivity aspartates and glutamates, new methods are in high demand. We report a technique inspired by the HaloTag technology, which employs nucleophilic substitution at chloroalkane-functionalised ligands by a specific aspartate residue. Embedding of alkyl bromide warheads into non-covalent inhibitors enables covalent modification of a glutamate in the lipoprotein binding chaperone - phosphodiesterase of retinal rod subunit delta (PDEδ), which shuttles prenylated lipoproteins between cellular membranes and thereby mediates their activity. Its hydrophobic ligand-binding pocket contains p.E88 as the only accessible nucleophile for covalent targeting. We show that a covalent inhibitor, termed DeltaTag, overcomes limitations of non-covalent inhibitors. DeltaTag labels PDEδ at its p.E88 under biologically relevant conditions, modulates mammalian target of rapamycin (mTOR) signalling by disrupting the PDEδ-Rheb (Ras homologue enriched in brain)-mTORC1 (mTOR complex 1) axis and inhibits cancer cell proliferation. This proof-of-concept study demonstrates that the design strategy holds promise for the covalent modification of proteins with lipophilic binding sites that lack accessible reactive amino acids but contain specific carboxylates.

Targeted covalent inhibitors (TCIs) have gained increasing attention in chemical biology and drug discovery[1,2], and recently have given rise to a new generation of drugs, targeting, for instance, different kinases[3] and the KRas[G12C] protein[4,5]. A typical TCI comprises a covalent warhead incorporated into a reversible ligand, enabling covalent binding to the protein of interest[6]. The majority of electrophilic warheads target reactive cysteines, which occur at 2.3% abundance in the proteome[7,8]. In contrast, aspartic and glutamic acids (Asp and Glu) represent 12% of the entire proteome and are often of significance in retaining protein structure and function[9,10]. However, due to their intrinsically weak nucleophilicity under physiological conditions and the relatively lower stability of the usually formed ester linkage, these amino acids have been

subject to covalent inhibition in relatively few cases[7,8], and new methods for covalent Asp- and Glu-targeted inhibition are in high demand.

Most carboxylate targeting warheads, for example, ynamides, tetrazoles and diazocarboxamides, have been developed in the context of proteome profiling, but due to their limited reactivity and selectivity, TCIs based on these functional groups have rarely been developed (Fig. 1a)[7,8]. Alternative carboxylate labelling methods include the use of glycine ethyl ester (GEE, Fig. 1a) together with carbodiimide to tag surface-exposed Asp (D) and Glu (E) for mass spectrometry-based footprinting[11,12]. Among the reported TCIs targeting carboxylates, an epoxide fragment has been employed in an Asp-targeting glyceraldehyde-3-phosphate dehydrogenase (GAPDH)

[1]Department of Chemical Biology, Max Planck Institute of Molecular Physiology, Dortmund, Germany. [2]Crystallography and Biophysics Facility, Max Planck Institute of Molecular Physiology, Dortmund, Germany. [3]Lead Discovery Center GmbH, Dortmund, Germany. [4]Analytics Core Facility Essen (ACE), Chemical Biology, Faculty of Biology, University of Duisburg-Essen, Essen, Germany. [5]Faculty of Chemistry and Chemical Biology, Technical University Dortmund, Dortmund, Germany. [6]These authors contributed equally: Ruirui Zhang, Jie Liu. ✉e-mail: Herbert.waldmann@mpi-dortmund.mpg.de

**a** Reported covalent warheads targeting Asp/Glu

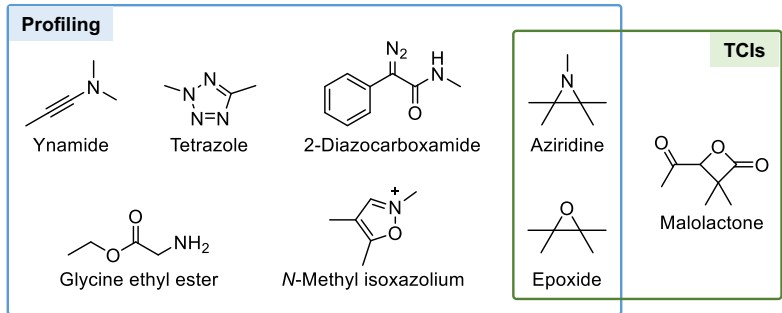

**b** Selected previously developed PDEδ inhibitors

**c** Inspiration for this work

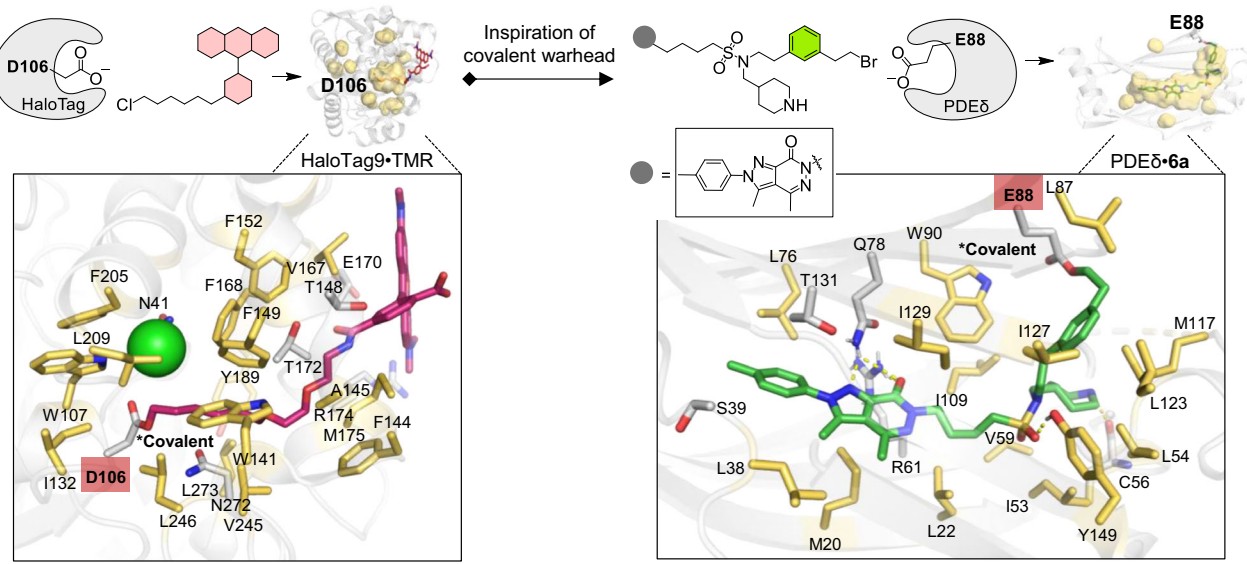

inhibitor reported by Uesugi et al.[13] and in a dual KRas^G12D/G12C inhibitor reported by Yu et al. (Fig. 1a)[14]. Shokat et al. have reported malolactone-based inhibitors that covalently target KRas^G12D (Fig. 1a)[15], and aziridine-based warheads (Fig. 1a) have been employed for covalent inhibition of KRas^G12D [16], as well as other targets such as

mitochondrial carrier homologue 2 (MTCH2), RUN and FYVE domain-containing protein 1 (RUFY1), and delta(24)-sterol reductase (DHCR24)[17].

We have recently shown that an isoxazolium salt derived from Woodward's reagent K (WRK) embedded in *bis*-sulphonamide

**Fig. 1 | Covalent modifications targeting glutamate and aspartate side chains.**
**a** Reported covalent warheads targeting Asp/Glu, for profiling and TCIs[13–18,67–72].
**b** Selected PDEδ inhibitor chemotypes developed previously, their reported binding affinities ($K_D$) and their co-crystal structures with PDEδ (PDB codes: 4JVF, 5E80, 5ML3 and 5NAL), shown with key interactions in the PDEδ prenyl binding pocket (hydrogen bond as yellow dotted line; π stacking as blue dotted line)[18,19,21,22].
**c** This work: development of covalent PDEδ inhibitors inspired by the HaloTag

ligand reaction mechanism. Crystal structures of a TMR ligand bound to HaloTag at its aspartic acid p.D106 (PDB code: 6ZVY[73]) and compound **6a** bound to PDEδ at its p.E88 (this work; PDB code: 9HMD) are shown. Cavities and hydrophobic pockets are indicated with yellow surfaces. In the closed-up view of the binding pockets for both crystal structures, Asp/Glu for covalent modification (p.D106 and p.E88) are highlighted in red boxes. Hydrophobic amino acid residues in the protein binding sites are highlighted as yellow sticks. The green sphere represents a chloride ion.

inhibitors enables covalent targeting of p.E88 in the binding pocket of phosphodiesterase of retinal rod subunit delta (PDEδ) (Fig. 1b), which does not contain a reactive cysteine or lysine accessible for covalent targeting[18,19]. The lipoprotein binding chaperone PDEδ binds the prenylated termini of several GTPases, including the Ras, Rab, Rheb and Rho proteins[20]. PDEδ establishes the dynamic intracellular localisation of these cargos, for instance, of the Ras[19,21,22] and Rheb proteins[23], and thereby plays an essential role in the regulation of their membrane localisation and proper function. We and others[19,21,22,24–26] have reported several potent ($K_D$ values in the low nanomolar and picomolar range), reversible inhibitors of PDEδ (Fig. 1b)[19,21,22] but their efficacy is intrinsically limited by the counteracting Arl2/3 GTPases, which upon allosteric binding stabilise an 'open' form of PDEδ such that even high-affinity ligands have an increased off-rate and are released from the lipoprotein binding site[23]. We envisaged that TCIs of PDEδ would overcome ligand release by Arl2/3; however, the isoxazolium warhead mentioned above is unstable under physiological conditions and shows higher reactivity towards Cys in biochemical assays, restricting its application[18].

We now describe a covalent modification approach employing biocompatible warheads with adequate and balanced reactivity and selectivity for targeting PDEδ at p.E88 in particular. To this end, we have drawn inspiration from the HaloTag technology. Innate to the HaloTag system is a covalent conjugation reaction between ligands with a reactive chloroalkane linker and a specific aspartic acid (D106) deep in the hydrophobic pocket of the HaloTag protein, forming a stable ester construct (Fig. 1c)[27]. Although in PDEδ the glutamate residue p.E88 is not located deep in, but rather at the upper end of the prenyl binding site, we reasoned that in both cases the target carboxylate residues are embedded in a similarly hydrophobic environment and the formed esters would remain relatively inaccessible to hydrolysis (Fig. 1c). The Asp-targeting reaction in the HaloTag protein occurs in high rate with low background labelling of other proteins[27]. In light of these arguments, we explored the stability, reactivity, and specificity of haloalkane substituents in covalent modification of PDEδ at p.E88.

Here, we demonstrate the feasibility of employing alkyl bromides derived from reversible inhibitors for targeted covalent modification of PDEδ at its binding site p.E88 under biologically relevant conditions. The best covalent inhibitor, termed DeltaTag, overcomes the limitations of reversible inhibitors. DeltaTag effectively labels PDEδ in cells, modulates signal transduction through the mTOR pathway in which the PDEδ cargo GTPase Rheb holds a prominent position, and impairs proliferation of human cancer cells with improved potency beyond reversible inhibitors. These findings may open up opportunities for the treatment of diseases involving PDEδ or its downstream effectors implicated, for instance, in oncogenesis. With a proof-of-concept study in the model system of PDEδ, covalent warheads inspired by the HaloTag system may also be applicable to other proteins with specific carboxylic acid residues embedded in hydrophobic binding sites, such as the lipoprotein chaperone UNC119.

## Results

### A hybrid strategy for covalent PDEδ inhibitor development
For covalent inhibitor development, the amide side chain of the nanomolar PDEδ inhibitor Deltazinone **1** (Fig. 2a)[22] was replaced by different alkyl halides (**2a**–**d**, Fig. 2a, and Supplementary Information,

see Supplementary Fig. S1 for details and for structures and numbering of all compounds **2**–**5**). Covalent attachment of the compounds to PDEδ was analysed by means of MALDI mass spectrometry, and the degree of covalent modification was estimated from the relative intensity of the signals recorded for the formed adducts (Fig. 2b). However, compounds **2a**–**d** did not efficiently label the protein. Only **2d** bearing a pentyl bromide warhead modified the target to a limited extent of $8 \pm 5\%$ after 24 h (Supplementary Fig. S1). While this finding demonstrated the feasibility of the approach, it also called for structural optimisation and improvement of affinity to PDEδ. Comparison of the crystal structures of inhibitors Deltazinone and the more potent Deltasonamide revealed that a hydrogen bond between the carbonyl group of Cys56 (C56) of PDEδ and a piperidine embedded in Deltarasin is important for higher potency (Fig. 1b). We hence employed a fragment-based hybrid design strategy[28] and introduced a piperidine into the amide side chain to anchor the ligand in the PDEδ pocket and to correctly position the warhead towards p.E88 for covalent bond formation (**3a**–**d**, Fig. 2a). Alkyl chlorides **3a** ($n = 1$, X = Cl), **3b** ($n = 3$, X = Cl) and alkyl bromide **3c** ($n = 1$, X = Br, Fig. 2a) were not reactive enough to yield any detectable covalent adduct after 24 h while long chain alkyl bromide **3d** ($n = 3$, X = Br, Fig. 2a) covalently bound to the protein, albeit with limited efficiency ($11 \pm 4\%$ at 24 h; Supplementary Fig. S1). We noted that unsymmetrically substituted amides form rotamers due to slow rotation around the amide bond, and reasoned that in the binding pocket of PDEδ, one rotamer might have the reactive electrophile correctly positioned for reaction with the glutamic acid, while in the other rotamer, the warhead might point away from the nucleophile, ultimately resulting in limited covalent binding. Bioisosteric replacement of amides by sulphonamides[29] overcame this limitation. In compounds **4a**–**d** (Fig. 2a), free rotation around the sulphonamide bond is possible, enabling correct orientation of the reactive warhead towards glutamic acid p.E88, and thereby, more efficient covalent labelling. Variation of the linker between the sulphonamide and the pyrazolopyridazinone fragment revealed that four methylene units are most favourable, with **4b** showing the highest degree of covalent binding ($23 \pm 3\%$ after 24 h, Fig. 2a, b). Subsequent variation of the alkyl halide chain (**5a**–**h**, Supplementary Fig. S1) revealed that alkyl chlorides with up to seven carbon atoms did not display any covalent modification of PDEδ over 24 h (Fig. 2a, b; Supplementary Fig. S1) whereas alkyl bromides moderately labelled the protein (up to $62 \pm 10\%$ with increasing alkyl chain length from C-5 to C-7; **4b**, **5d**, **e**, Fig. 2a, b). Further incorporation of polar oxygen atoms in the alkyl chain was not beneficial for covalent bond formation (**5i-l**, Supplementary Fig. S1). Gratifyingly, introduction of a conformational constraint into the C-7 alkyl bromide chain to thermodynamically favour binding by reducing entropy loss resulted in prototype covalent inhibitor **6a**, which displayed a labelling efficiency of $95 \pm 2\%$ after 24 h incubation (Fig. 2a, b; estimated apparent second-order rate constant $k_{inact}/K_I = 31\,M^{-1}s^{-1}$, Supplementary Fig. S1c).

Exposure of compounds **5e** and **6a** to different aqueous buffers with varying pH values and also in the presence of excess glutathione (GSH) revealed that the alkyl halides are reasonably stable in aqueous solution (half-life $t_{1/2}$ in HEPES buffer at pH 7.5 = 85 h and 480 h, respectively, Supplementary Fig. S2) and react only slowly with the sulphur nucleophile (half-life for GSH adduct formation with a 10-fold excess of GSH at 10 mM, $t_{1/2}$ GSH = 4332 h and 1035 h, respectively, Supplementary Fig. S2). In addition, alkyl bromides **5e** and **6a** have

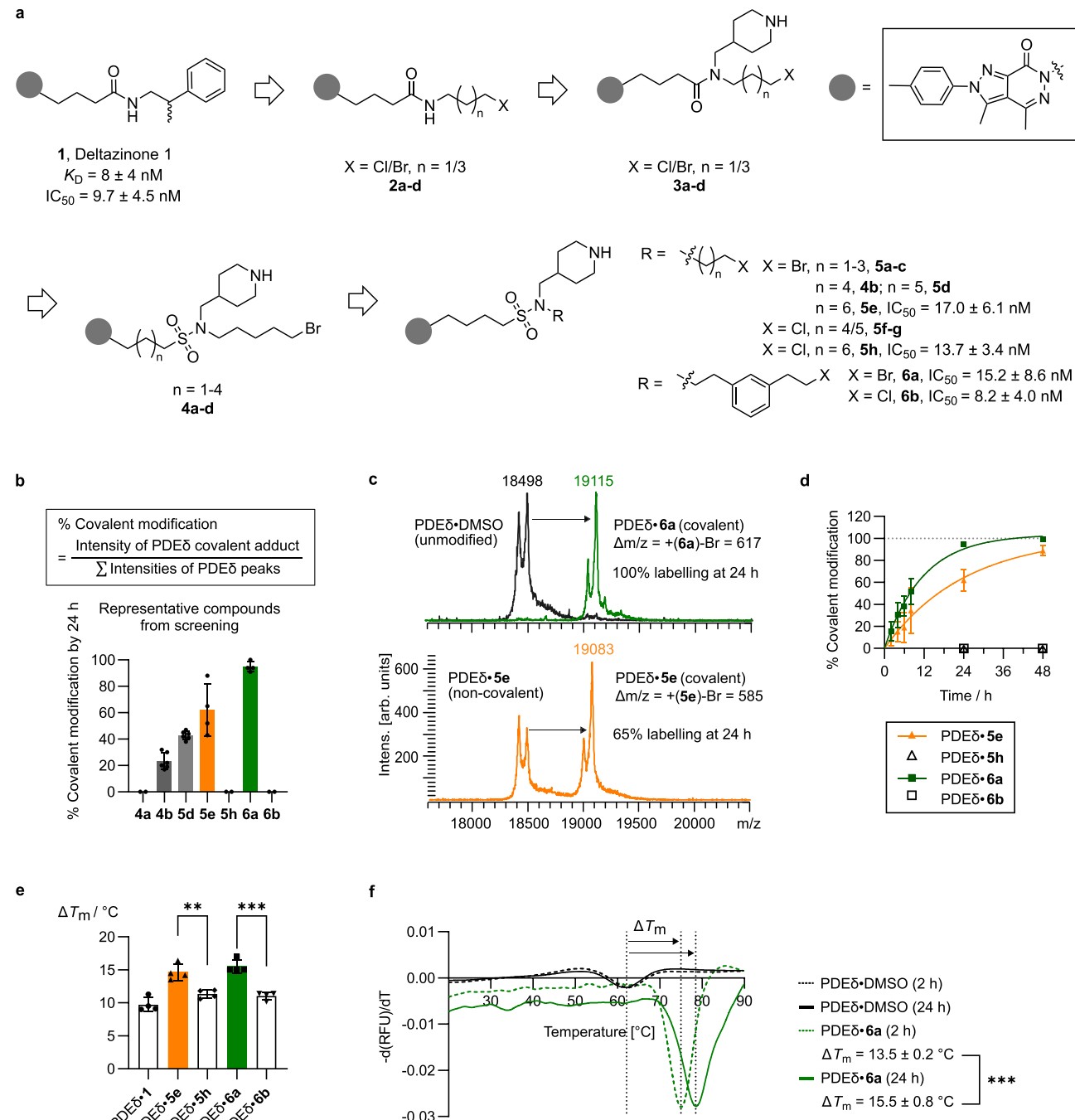

**Fig. 2 | Structural development of PDEδ inhibitors. a** Structures and binding affinities ($IC_{50}$ determined by competitive fluorescence polarisation assay, see Supplementary Fig. S3) of key Deltazinone-based compounds **1**-**6**. **b** % Covalent modification of representative compounds at 24 h as determined by MALDI mass spectrometry. PDEδ (20 μM) was incubated with compounds (60 μM, 0.6% DMSO) in HEPES buffer (20 mM HEPES, 150 mM NaCl, pH = 7.5) at 37 °C for 24 h before MALDI analysis. Percentages of covalent adduct formation were estimated by the relative intensity (arbitrary units, arb.units) of the respective peaks in MALDI spectra. Data are presented as mean ± s.e.m., representative of biological replicates with different stocks of protein solutions ($n$ = 6 for **4b** and **5d**, $n$ = 4 for **5e** and **6a**, $n$ = 2 for negative results). **c** Positive ion-mode MALDI-TOP mass spectra of PDEδ in the presence or absence of **5e** or **6a** after 24 h of incubation in HEPES buffer at 37 °C, representative of four independent experiments ($n$ = 4). **d** Kinetics of covalent modification of PDEδ (20 μM) by **5e** and **6a** (60 μM). The corresponding chloro-

substituted **5h** and **6b** are reversible inhibitors of PDEδ. Data are presented as mean ± s.e.m. (biological replicates $n$ = 4). **e** Melting temperature shift ($\Delta T_m$) of compound-bound PDEδ as compared to unmodified PDEδ determined by NanoDSF. Data points represent individual biological replicates. Data are presented as mean ± s.d. (biological replicates $n$ = 4). Unpaired $t$-test, PDEδ•**5e** versus PDEδ•**5 h**, two-tailed $p$-value = 0.0031; PDEδ•**6a** versus PDEδ•**6b**, two-tailed $p$-value = 0.0002. **f** NanoDSF of PDEδ in the presence or absence of **6a** after 2 h and 24 h incubation. The $y$-axis shows the first derivative of the relative fluorescence ratio (RFU) at wavelength 330 nm/350 nm over temperature (T), as a measure of the rate of protein unfolding. Melting temperatures were recorded at the inflexion points. Melting temperature shifts ($\Delta T_m$) were presented as mean ± s.d., representative of six independent experiments ($n$ = 6). Unpaired $t$-test, two-tailed $p$-value = 0.0001. Source data are provided as a Source Data file.

relatively low intrinsic reactivity to carboxylic acids in aqueous solution (half-life $t_{1/2}$ in acetate buffer, pH 5.6 = 43 h and 145 h, respectively, Supplementary Fig. S2). Compound **6a** showed faster covalent binding to PDEδ (Fig. 2d) and was consistently more stable than compound **5e** under all aqueous conditions investigated. Covalent modification of PDEδ by **6a** in different solvents is pH-independent with no significant changes in labelling efficiency (Supplementary Fig. S1d). The formation of the ester adduct also appeared to be irreversible, since we did not observe any reversal to unmodified protein after incubation of the formed adduct in these solvents for 3 days at 37 °C. This finding is in agreement with the previously reported stability of an ester bond formed with PDEδ in the presence of a 50-fold excess of hydroxylamine and release factor Arl2[18].

Compounds **5e** and **6a** were selected for further investigation as covalent PDEδ inhibitors, and structurally closely related but reversible chloro-substituted analogues **5h** and **6b** were included for comparison. Similar to Deltazinone **1**, these compounds exhibit nanomolar binding affinity to PDEδ (Fig. 2a, $IC_{50}$ values determined by competitive fluorescence assay, see Supplementary Fig. S3 for details) and upon binding stabilised the protein with shifts in melting temperature ($\Delta T_m$) > 10 °C (Fig. 2e). For the covalently modified PDEδ adducts thermal stabilisation was higher than for their reversible counterparts (PDEδ•**5e**, $\Delta T_m = 14.7 \pm 1.2$ °C versus PDEδ•**5h**, $\Delta T_m = 11.3 \pm 0.6$ °C, unpair $t$-test, $p$-value = 0.0031; and PDEδ•**6a**, $\Delta T_m = 15.5 \pm 1.0$ °C versus PDEδ•**6b**, $\Delta T_m = 11.1 \pm 0.6$ °C, unpaired $t$-test, $p$-value = 0.0002, respectively). Compounds **5e** and **6a** covalently bound to PDEδ in a temperature-dependent (Supplementary Fig. S3a) and time-dependent manner (Fig. 2d). With time, the degree of covalent binding of **6a** to PDEδ increased from $16 \pm 9\%$ at 2 h to $95 \pm 2\%$ at 24 h (Fig. 2d), which was reflected in the enhancement of thermostability from $\Delta T_m = 13.5 \pm 0.2$ °C to $\Delta T_m = 15.7 \pm 0.8$ °C at the respective time points (unpaired $t$-test, $p$-value = 0.0001, Fig. 2f).

### Selective covalent modification of PDEδ at glutamic acid p.E88

In order to characterise the observed covalent modification of PDEδ by compounds **5e** and **6a**, we analysed the covalent PDEδ•**5e** and PDEδ•**6a** adducts with mass spectrometric evaluation of the peptides formed by Glu-C digestion. The modified peptide sequence QKVQKVYFKGQCLEE corresponding to the exact mass was identified, based on a good sequence coverage of PDEδ (Fig. 3a, see Supplementary Information, Supplementary Table 1 for details). The results indicated that compounds **5e** and **6a** covalently bound to PDEδ either at p.E88 or p.E89, with a higher probability for modification at p.E88 (Supplementary Table 1). Given that the compounds were added in excess to PDEδ (10:1 compound to protein ratio) for incubation and that PDEδ has other potential nucleophilic sites for covalent binding (we searched in parallel for modifications at other Glu, Asp, Cys, Thr, Ser, Tyr, Lys and His residues)[18], the fact that we did not observe other modified peptides provides strong support for selectivity of the covalent labelling of PDEδ at its binding site p.E88 by **5e** and **6a**, even when other sites may have been more reactive and/or more accessible.

We obtained crystal structures for both adducts PDEδ•**5e** and PDEδ•**6a**, which unambiguously confirmed that the compounds covalently bound to PDEδ within its binding pocket at p.E88 (Fig. 3b), with well-defined electron density for the covalent ester bond between p.E88 and the compounds (Fig. 3c, d). Analysis of the crystal structures of PDEδ•**5e** and PDEδ•**6a** indicated that both ligands are buried in the hydrophobic prenyl binding site of PDEδ. Comparison with the structure of a Deltazinone derivative bound to PDEδ (PDB code: 5E80) showed that both **5e** and **6a** are located even deeper in the pocket[22]. The positioning of the compounds within the PDEδ binding pocket and the orientation of the covalent warheads therefore explain the preferential reactivity towards p.E88, as observed in the mass spectrometric analysis of the covalent adducts. Even though the MS method

could not distinguish between modifications at p.E88 and p.E89, labelling at p.E88 was the only plausible mode with the compounds occupying the binding pocket, as p.E89 has opposite orientation and is solvent-exposed (Fig. 3c, d). Currently, p.E88 is also the only accessible nucleophilic residue in PDEδ reported for covalent targeting[18]. For covalent inhibitors adopting similar positioning and warhead orientation as **5e** and **6a**, no other nucleophilic residues in spatial proximity exhibited covalent reactivity, thereby demonstrating in vitro labelling selectivity. Besides mostly hydrophobic interactions, the ligands are held in the pocket by hydrogen bonds with Arg61 (R61), Gln78 (Q78) and Tyr149 (Y149) by analogy to the binding of Deltazinone **1** (Fig. 3e, f). Different from a C-3 linker that was employed in Deltazinone **1** and optimised for establishing the hydrogen bond between the amide and Y149[22], an additional methylene unit in the linker is necessary to position the sulphonamide in the vicinity of Y149 for an H-bond with a distance of 2.1 Å and 2.0 Å in structures with **5e** and **6a**, respectively. As designed following the fragment-based hybrid strategy, the piperidine ring in PDEδ•**5e** forms an additional hydrogen bond (2.2 Å) with the backbone carbonyl of p.C56, which hence anchors the ligands and orients the reactive warhead towards p.E88 to form an ester bond (Fig. 3e). In the structure of PDEδ•**6a**, this H-bond is weakened as the piperidine adopts a different conformation with the hydrogen pointing away from p.C56 in a distance of 3.1 Å (Fig. 3f). However, this arrangement not necessarily compromises the significance of this hydrogen bond in correctly positioning the ligand at its 'reversible' binding stage to allow rapid formation of the covalent linkage with p.E88. These key interactions hold the ligands in very favourable positions, and their orientation in the PDEδ pocket directly correlates to their covalent binding efficiency. We termed the prototype compound **6a** with its warhead inspired by HaloTag ligands and of the highest labelling efficiency to PDEδ as DeltaTag.

### Covalent modification of PDEδ by DeltaTag in the presence of Arl2

Under physiological conditions, the efficacy of reversible PDEδ inhibition is inherently limited by the Arl2/3 GTPase release factors, which allosterically bind to PDEδ and stabilise its 'open' state that counteracts inhibitor binding[23]. We therefore monitored the in vitro kinetics of covalent PDEδ modification by DeltaTag (**6a**) in the presence of Arl2•GppNHp and observed that the reaction proceeded at a rate comparable rate to the rate in the absence of Arl2•GppNHp (PDEδ: Arl2•GppNHp = 1:1, 3 eq. of **6a**, Fig. 4a). This finding suggests that binding of DeltaTag (**6a**) to PDEδ is not significantly disrupted through release by Arl2. Formation of the ester adduct at 37 °C shifts the reaction equilibrium without notable reduction in reaction rate, especially when the electrophile is present in excess. We further modified the fluorescence polarisation assay previously employed to determine whether a covalent inhibitor would behave differently from a reversible inhibitor in the presence of Arl2. Orienting model validation suggested that no covalent bond was formed at room temperature upon incubation with a 12.5-fold excess of DeltaTag under the assay conditions (Supplementary Fig. S3a), while release by Arl2•GppNHp was confirmed by the apparent increase in $K_d$ of the fluorescently labelled PDEδ ligand atorvastatin (FA probe) to PDEδ (Supplementary Fig. S3b). At the 'reversible' stage, DeltaTag (**6a**) and Deltazinone **1**, which have comparable apparent binding affinity to PDEδ ($IC_{50} = 15.2 \pm 8.6$ nM versus $IC_{50} = 9.7 \pm 4.5$ nM, Supplementary Fig. S3c), behave similarly in the presence of Arl2•GppNHp (Fig. 4b), suggesting that both could be released by Arl2•GppNHp. At 37 °C, covalent attachment of DeltaTag to PDEδ, in principle, avoids inhibitor release through allosteric binding of Arl2, given the apparent irreversibility of ester bond formation inside the binding pocket. In the binding site, the ester is relatively inaccessible to hydrolysis. The formation of a covalent bond to PDEδ led to an approximately 2-fold increase in the apparent binding affinity of DeltaTag (**6a**) measured in

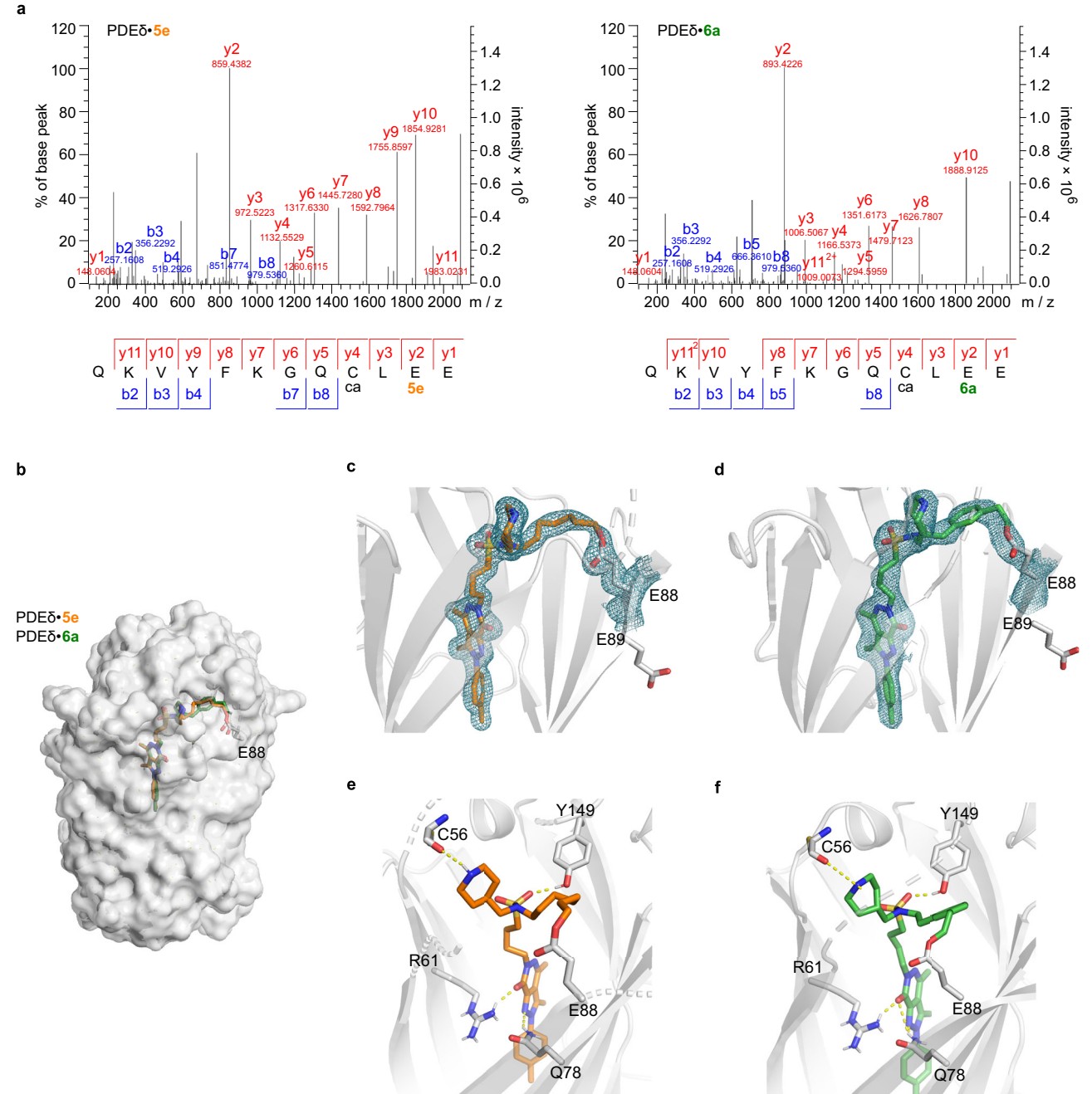

**Fig. 3 | Compounds 5e and 6a covalently bind to PDEδ at p.E88. a** LC-MS/MS spectra of compound **5e**- or **6a**-modified PDEδ peptide (amino acid 78-89), each representative of three biological replicates (*n* = 3, see Supplementary Table 1 for details). The sequence and detected fragments of the peptide are shown. Sequence-matched peaks are labelled in red and blue. Identified modifications at the amino acids are indicated below the peptide sequence—"ca" stands for carbamido-methylation of cysteine, and the best identified compound modification is localised at E88 for both compounds. **b** Superimposition of X-ray co-crystal structures of covalent adducts, PDEδ•**5e** (1.7 Å, PDB code: 9HMC) and PDEδ•**6a** (2.5 Å, PDB code: 9HMD). **c** $2F_o–F_c$ map for ligands **5e** and p.E88 in cyan mesh ($\sigma$ = 1.0). Solvent-

exposed p.E89 is also shown with the side chain. **d** $2F_o–F_c$ map for ligands **6a** and E88 in cyan mesh ($\sigma$ = 1.0). Solvent-exposed p.E89 is also shown with the side chain. **e**, **f** Key hydrogen bonding interactions (yellow dotted line) of ligand **5e** (orange) and **6a** (green) to PDEδ, respectively. The interacting amino acids, Y149, R61 and Q78 are highlighted and shown with the side chains. The backbone carbonyl of C56 forms another hydrogen bond to the piperidine of the ligands. See Supplementary Table 5 for details of X-ray crystallography data collection and refinement statistics. See Supplementary Fig. S9 for images of electron density maps and omit maps for the crystal structures.

the competitive displacement titration assay ($IC_{50}$ = 6.5 ± 3.5 nM at 37 °C, Supplementary Fig. S3d). For the reversible inhibitor Deltazinone **1**, a similar binding affinity was recorded with the change in temperature ($IC_{50}$ = 13.3 ± 8.7 nM at 37 °C, Supplementary Fig. S3d). At 37 °C, covalent inhibitor DeltaTag (**6a**) was, therefore, distinguished from the reversible Deltazinone **1** in the presence of Arl2•GppNHp

($IC_{50}$ = 10.1 ± 2.6 nM versus $IC_{50}$ = 31.0 ± 12.7 nM, Fig. 4c). When Delta-Tag (**6a**) was pre-incubated with PDEδ at 37 °C overnight before addition of Arl2•GppNHp, we observed an additional 2-fold increase in the apparent binding affinity ($IC_{50}$ = 4.8 ± 1.5 nM, Fig. 4c), suggesting that the change of PDEδ conformation induced by Arl2 allosteric binding negatively impacts DeltaTag binding.

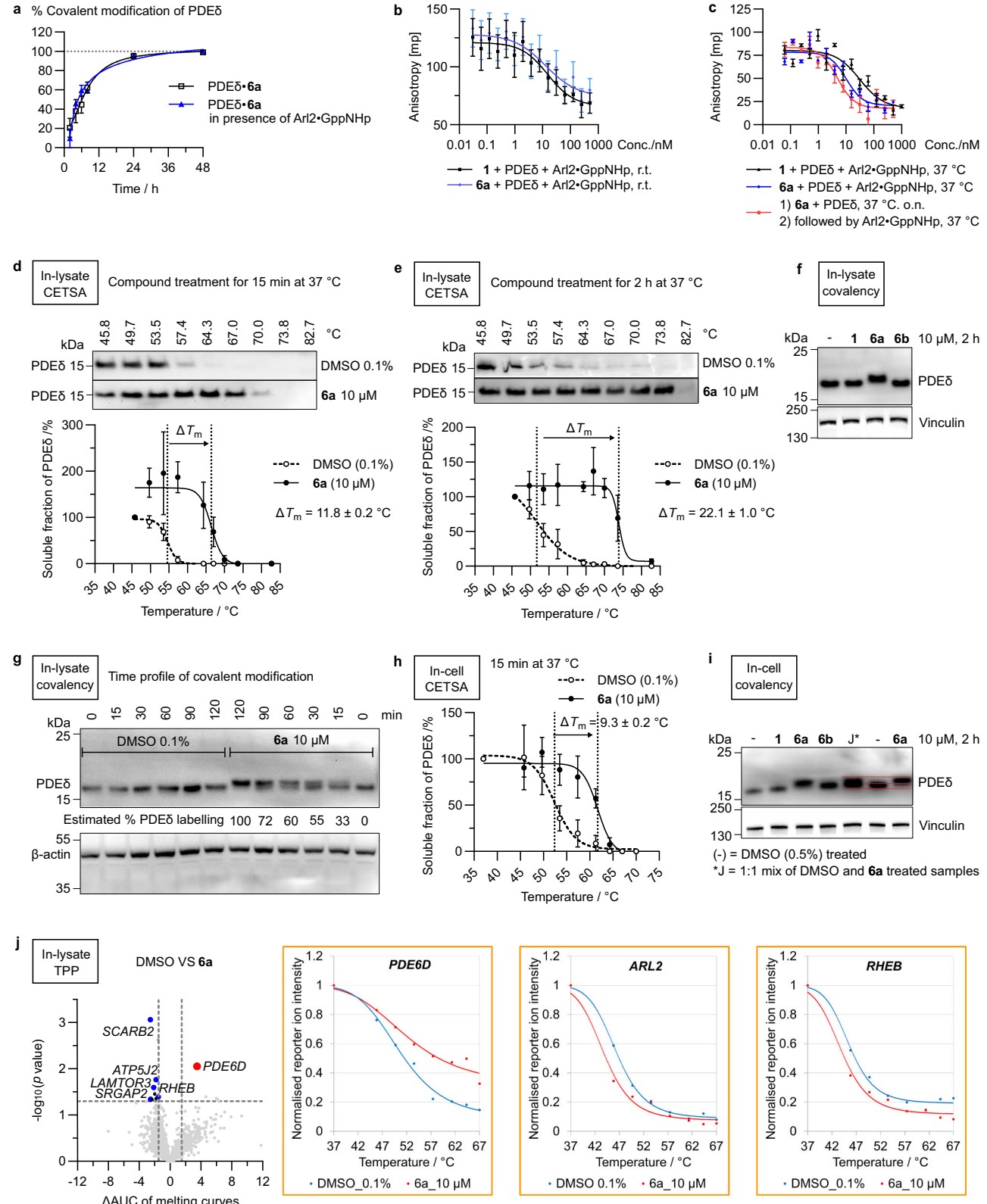

Transitioning from in vitro models to cellular systems, target engagement of PDEδ by DeltaTag (**6a**) in the complex cellular proteome was first investigated by means of cellular thermal shift assays (CETSA). After initial treatment with DMSO or DeltaTag (**6a**) at 10 μM for 15 min at 37 °C, Jurkat cell lysates were subjected to heating with a temperature gradient with the melting behaviour of PDEδ visualised by Western blot-based analysis (Fig. 4d). In this experiment PDEδ was stabilised by DeltaTag (**6a**), with a melting temperature shift ($\Delta T_m$) of 11.8 ± 0.2 °C (Fig. 4d). When the lysate was incubated with DeltaTag (**6a**) for 2 h at 37 °C, there was an enhanced stabilisation of PDEδ resulting in a $\Delta T_m$ of 22.1 ± 1.0 °C (Fig. 4e), consistent with a time-dependent covalent modification by DeltaTag (**6a**). To further validate the covalent nature of this modification of PDEδ by **6a**, we visualised the increase in molecular weight on the anti-PDEδ immunoblot. A small

**Fig. 4 | DeltaTag binding to PDEδ in the presence of Arl2. a** Kinetics of covalent modification of PDEδ (20 μM) by **6a** (60 μM) in the presence of Arl2•GppNHp (20 μM). Data are presented as mean ± s.e.m., representative of biological replicates $n = 3$. **b, c** Competitive fluorescence polarisation assays in the presence of Arl2•GppNHp at r.t. (**b**) and 37 °C (**c**). The respective compound was incubated with PDEδ (40 nM), Arl2•GppNHp (40 nM) and FITC-labelled atorvastatin (FA-probe, 24 nM). Data are presented as mean ± s.e.m. ($n = 4$ for **6a** at r.t., $n = 3$ for the rest). **d, e** In-lysate cellular thermal shift assay (CETSA) with incubation for 15 min (**d**) and 2 h (**e**). Data are presented as mean ± s.e.m., representative of biological replicates $n = 3$. **f** In-lysate covalent modification of PDEδ at 2 h, representative of biological replicates $n = 3$. **g** Time profile of in-lysate covalent modification of PDEδ by **6a**, representative of biological replicates $n = 3$. **h** In-cell CETSA (15 min incubation). Data are presented as mean ± s.e.m., representative of biological replicates $n = 3$. **i** In-cell covalent modification of PDEδ at 2 h, representative of biological replicates $n = 3$. **j** Thermal proteome profiling (TPP). Volcano plot with $-\log_{10}$ ($p$-value) against change in area under the melting curves (ΔAUC) is shown. Two-tailed $p$-values were determined by applying a paired $t$-test comparing the AUC of melting curves between DMSO-treated and compound-treated conditions, from biological replicates $n = 3$. Significance cut-offs: $p$-value ≤ 0.05, ΔAUC ≤ -1.5 or ≥1.5. Grey dots above threshold lines: false positive hits with no good fitting or inconsistent melting. Red dot: PDEδ (*PDE6D*) as the only significantly stabilised target. Blue dots: destabilised hits functionally linked to PDEδ. Representative fitted melting curves (PDEδ, Arl2 and Rheb) are shown, with DMSO controls in blue and compound-treated conditions in red. Orange frames: protein identification by at least two razor and unique peptides and a reporter ion intensity ≥ 1 × 10⁴ for the first three temperatures. Source data are provided as a Source Data file.

but clear upward shift of the band indicates covalent binding of DeltaTag (**6a**) to PDEδ upon treatment at 10 μM for 2 h, as compared to treatment with reversible PDEδ inhibitors Deltazinone **1** and **6b** (Fig. 4f). The in-lysate covalent labelling kinetics further supports the time-dependent nature of covalent modification and an estimated full labelling of PDEδ in 50 μg cell lysates by 10 μM DeltaTag within 2 h (Fig. 4g). In-cell CETSA also suggests rapid PDEδ engagement with a comparable thermal shift $\Delta T_m$ of 9.3 ± 0.2 °C (Fig. 4h). In-cell labelling further supports covalent binding of DeltaTag (**6a**) to PDEδ within 2 h despite the counteracting inhibitor release by Arl2 under physiological conditions (Fig. 4i).

We further monitored the dynamic changes in the complex proteome upon DeltaTag treatment with thermal proteome profiling (TPP) (Fig. 4j). Together with the in-lysate CETSA, this mass spectrometry-based readout enabled assessment of proteome-wide changes in protein stability upon rapid target engagement after 15 min incubation at 37 °C[30]. In total, we identified 3903 proteins from TPP, of which 3048 were identified with at least two unique peptides in all three replicates. Analysis of the fitted melting curves revealed PDEδ as the only significant hit ($p ≤ 0.05$) that showed target stabilisation with the largest change in area under the curves (ΔAUC) as compared to treatment with DMSO (ΔAUC = 3.5, $p$-value = 0.008, Fig. 4j). Within the temperature gradient set by TPP (37–67 °C), PDEδ was the only protein identified with incomplete melting (Fig. 4j)—a behaviour that we observed in PDEδ inhibitor discovery before[18,19]. Compared to the melting temperature shift of PDEδ ($\Delta T_m$ = 11.8 ± 0.2 °C, Fig. 4d) validated by the parallel in-lysate CETSA with additional thermal treatment beyond 67 °C, other potential hits from the TPP experiment ($\Delta T_m$ at least ± 2 °C) displayed much weaker stabilisation (Supplementary Table 2), confirming PDEδ as an unequivocal cellular target of DeltaTag (**6a**).

Nine proteins were destabilised (ΔAUC ≤ -1.5, $p ≤ 0.05$), albeit with small changes in AUC (Fig. 4j, see Supplementary Information, Supplementary Table 2 for details). Although Arl2 was not among the significant hits, we observed destabilisation of this protein (ΔAUC = 0.97, $p$-value = 0.58, $\Delta T_m$ = −0.92 °C, Fig. 4j). This finding suggests that DeltaTag binding was not prevented through release by Arl2, and that the formation of the covalent adduct (approximately 33%, Fig. 4g) locked PDEδ in its 'closed' form, which adversely impacts allosteric binding of Arl2. Among the significant hits, Rheb GTPase (*RHEB*, ΔAUC = −1.5, $p$-value = 0.04, $\Delta T_m$ = −0.89 °C, Fig. 4j) is a known PDEδ cargo[20,23] and the observed change in its thermal stability may be secondary to competitive binding of the small molecule to PDEδ. Ligand-binding can also affect thermal stability of the regulatory units of protein complexes[30], which was observed with SLIT-ROBO Rho GTPase-activating protein 2 C (*SRGAP2*, ΔAUC = −2.5, Fig. 5a), which interacts with the PDEδ cargo Rho GTPase[20], as well as with Ragulator complex protein LAMTOR3 (*LAMTOR3*, ΔAUC = −2.1, Fig. 5a), which binds to Rag GTPase and mTORC1 in a signalling complex downstream of the PDEδ cargo protein Rheb. The dynamic changes in protein stability revealed by the TPP experiment supported on-target engagement of PDEδ by DeltaTag.

## Cellular effects of DeltaTag impact the PDEδ-Rheb-mTORC1 axis

The disruption of the interaction between PDEδ and farnesylated Rheb by DeltaTag was visualised by fluorescence lifetime imaging microscopy (FLIM) in living cells. To this end, HEK293T cells were transiently co-transfected to overexpress mCitrine-Rheb and mCherry-PDEδ (1:1.2 ratio of cDNA) such that mCitrine-Rheb exists mostly in the bound state with PDEδ. Distinct from the clustered fluorescence pattern observed in cells transfected only with mCitrine-Rheb, the cellular distribution of mCitrine-Rheb in the presence of mCherry-PDEδ was homogeneous throughout the cytoplasm (Fig. 5a), indicating a clear solubilisation of mCitrine-Rheb by its chaperone mCherry-PDEδ. When bound to a fluorescence acceptor, mCitrine-Rheb exhibited an average fluorescence lifetime of τ = 2.58 ± 0.04 ns, which was distinguished from its unbound state in the donor-only system with a longer lifetime of τ = 3.01 ± 0.02 ns (Fig. 5a, b). Within 10 min of 5 μM DeltaTag treatment under live-cell settings, the disruption of mCitrine-Rheb binding was evident by a noticeable redistribution of mCitrine-Rheb localisation to a clustered pattern to endomembranes and a significant increase in mCitrine lifetime from the untreated condition to τ = 2.81 ± 0.05 ns (Fig. 5a, b, unpaired $t$-test, $p$-value = 0.0036). Under endogenous conditions, overnight treatment of PA-TU-8902 cells with 5 μM DeltaTag led to a visible and sustained redistribution of Rheb to endomembranes, whereas in vehicle-treated and reversible Deltazinone **1**-treated cells, a more uniform pattern of the protein indicates solubilisation of Rheb throughout the cytoplasm and a loss of inhibition for PDEδ-Rheb interaction by the reversible Deltazinone **1** over a longer timeframe (Fig. 5c).

We further investigated the dynamic changes in cell signalling downstream of PDEδ-Rheb inhibition by phosphoproteomic profiling (Supplementary Fig. S4a). Consistent with the disruption of cytoplasmic Rheb localisation, where Rheb binds to the lysosomal membranes for proper signalling, we observed regulation of kinases related to suppression of the Rheb-mediated mTOR pathway upon treatment with **6a**, as shown by kinase-substrate enrichment analysis of the detected changes in phosphorylation[31–33] (Fig. 5d). Specifically, we observed reduced activity of the mammalian target of rapamycin (mTOR, encoded by *MTOR*) with a kinase $z$-score of −2.75, downregulation of casein kinase I (CK1, encoded by *CSNK1A1*) activity, which is a positive regulator of mTOR complexes 1/2 (mTORC1/2)[34] with a $z$-score of −2.95, and upregulation of a negative mTORC1 regulator, cAMP-dependent protein kinase (PKA, encoded by *PRKACA*)[35] with a $z$-score of +3.36 in the presence of **6a** (Fig. 5d). Further input of the significantly suppressed kinases into Reactome for pathway over-representation analysis[36] also confirmed the downregulation of the Rheb-mediated mTOR pathway upon DeltaTag treatment (Fig. 5e). Pathway overrepresentation of the set of downregulated kinases using GO against the KEGG database[37–39] revealed an additional list of significantly suppressed signalling pathways, many of which are directly involved in the PI3K-Akt-mTOR pathway (Supplementary Fig. S4b). The farnesylated Ras family GTPase Rheb is a positive regulator of mTORC1 activity. We further validated this inhibition of the mTOR pathway by

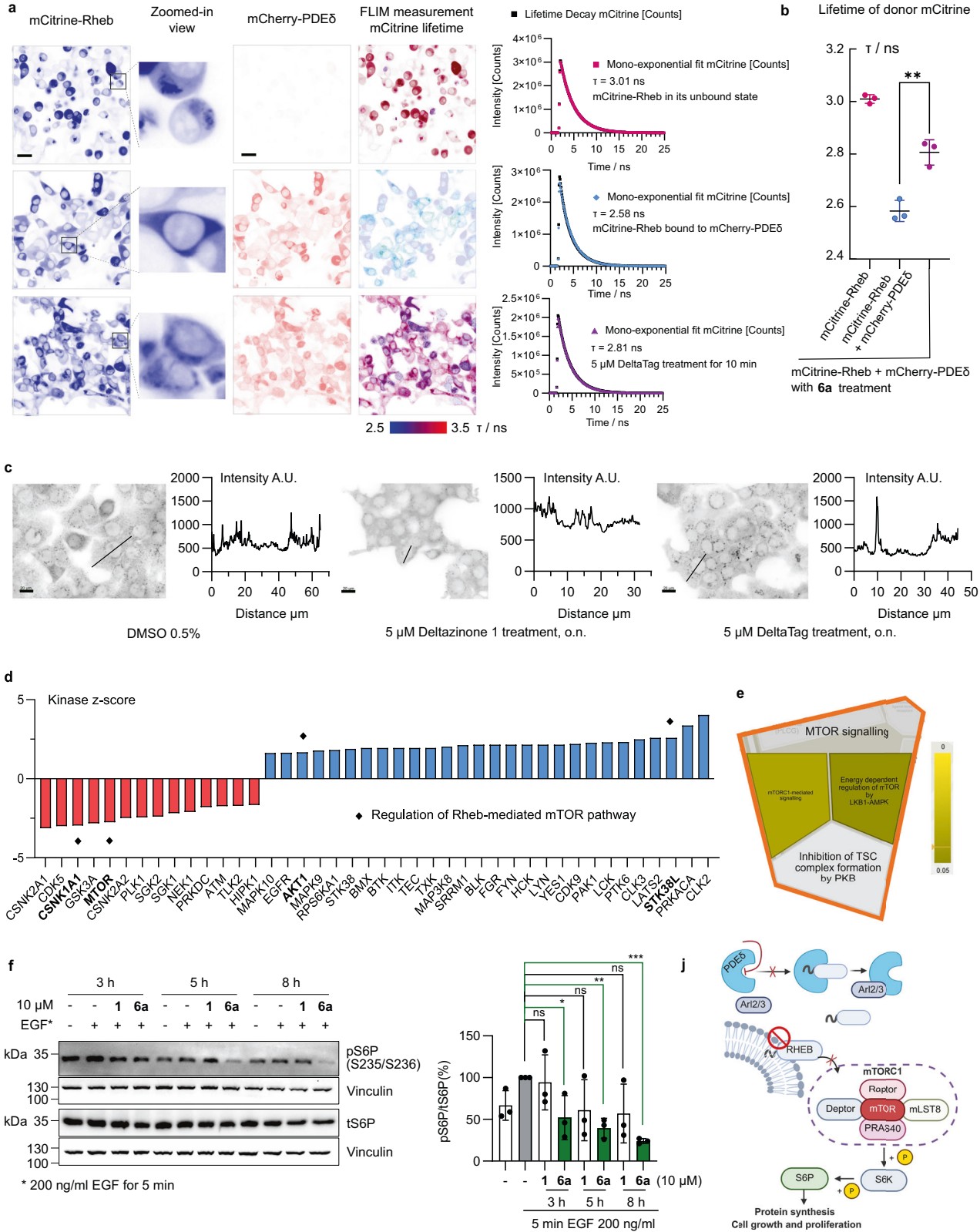

DeltaTag (**6a**) by examining phosphorylation of the mTORC1 substrate S6 ribosomal protein (S6P) in PA-TU-8902 cells by means of Western blot analysis (Fig. 5f). DeltaTag (**6a**) reduced S6P phosphorylation in a time-dependent manner and consistently showed higher activity than the reversible inhibitor Deltazinone **1** at 10 μM (Fig. 5f). The signalling signature with a reduction of mTORC1 activity (Fig. 5d, f) and an upregulation of AKT (Fig. 5d) which is catalysed by mTORC2 also

suggests a selective inhibition of the mTORC1 pathway coupled with a feedback loop[40].

## DeltaTag suppresses the growth of KRAS-mutant human cancer cells

Impairment of the PDEδ-Rheb-mTORC1 pathway by DeltaTag manifested itself in antiproliferative activity in human cancer cell lines. In the

**Fig. 5 | DeltaTag (6a) inhibits the PDEδ-Rheb-mTORC1 axis in cells. a** mCitrine lifetime (τ) measurement in HEK293T cells with mCitrine-Rheb only, or donor-acceptor system with mCitrine-Rheb bound to mCherry-PDEδ, or donor-acceptor system with DeltaTag treatment for 10 min. First column: fluorescence intensity distribution mCitrine-Rheb; second column: zoomed-in views for cellular mCitrine-Rheb distribution; third column: mCitrine-PDEδ; fourth column: global lifetime distribution; fifth column: lifetime decay of mCitrine and fitting with mono-exponential decay. Scale bar, 25 μm. Images represent biological replicates $n = 3$ (each with technical replicates $N = 6$). **b** Average lifetime of mCitrine. Data are presented as mean ± s.d., representative of biological replicates $n = 3$. Unpaired $t$-test, two-tailed $p$-value = 0.0036 (**). **c** Immunofluorescence staining of PA-TU-8902 cells with anti-Rheb antibody, overnight after treatment, representative of biological replicates $n = 3$. Intensity profiles (arbitrary unit, A.U.) along the lines are plotted against distance (μm). Scale bars, 20 μm. **d** Plot of kinase $z$-scores (two-tailed $p$-value ≤ 0.05) after kinase-substrate enrichment analysis (KSEA App with NetworKIN, substrate count cutoff = 2,

NetworKIN score cutoff = 1)[31-33] of significant hits from phosphoproteome profiling upon treatment of **6a** (see Supplementary Fig. S5 for details). **e** Reactome pathway overrepresentation[36] of downregulated kinases, with Voronoi visualisation zooming into mTOR signalling. The scale of colour intensity is an indication of the $p$-value of pathway overrepresentation. **f** Western blot analysis of S6P phosphorylation on Ser235 and Ser236 (S235/S236) and total S6P (tS6P) in PA-TU-8902 cells. (−/−) represents unstimulated DMSO-treated cells. Quantification of per cent pS6P/tS6P ± s.d. was normalised to EGF-stimulated DMSO control at each time-point, representative of biological replicates $n = 3$. Unpaired $t$-test, two-tailed $p$-values comparing each condition to EGF-stimulated DMSO control (−/+): 3 h: vs **1**, $p = 0.78$; vs **6a**, $p = 0.035$ (*); 5 h: vs **1**, $p = 0.138$; vs **6a**, $p = 0.0009$ (**); 8 h: vs **1**, $p = 0.102$; vs **6a**, $p < 0.0001$ (***); $p$-value > 0.05 are labelled as non-significant (ns). **j** Schematic representation of inhibition along the PDEδ-Rheb-mTORC1 axis, created in BioRender. Zhang, R. (2026) https://BioRender.com/dxvyoxr. Source data are provided as a Source Data file.

development of reversible inhibitor Deltazinone **1**, a correlation had been demonstrated between the inhibition of the cellular interaction of PDEδ and its client protein KRAS, and the antiproliferative activity of **1** in a panel of KRAS-dependent human cancer cell lines[22]. Therefore, by analogy, we investigated the antiproliferative activity of the newly developed inhibitors DeltaTag (**6a**), **6b** along with Deltazinone **1** in KRAS-dependent human carcinoma cell lines PA-TU-8902 (pancreatic), MIA PaCa-2 (pancreatic), NCI-H358 (lung), SW480 (colorectal), as well as KRAS wild-type-expressing BxPC-3 (pancreatic) cell line with real-time live-cell imaging by Incucyte (Fig. 6a). By analogy to Deltazinone **1**[22], DeltaTag (**6a**) has the strongest growth inhibition for the KRAS-dependent PA-TU-8902 cell line and the weakest growth suppression for the KRAS wild-type-expressing BxPC-3 cell line (Fig. 6a, b). DeltaTag (**6a**) exhibited a stronger growth inhibitory effect than Deltazinone **1**, except in PA-TU-8902 where both compounds showed comparable potency (IC$_{50}$ of **6a** = 6.8 ± 0.5 μM versus IC$_{50}$ of Deltazinone **1** = 4.6 ± 1.4 μM, Fig. 6b), and in NCI-H358 cells where Deltazinone **1** induces significantly stronger growth inhibition (IC$_{50}$ = 0.14 ± 0.11 μM, Fig. 6b). While comparable in IC$_{50}$ values in PA-TU-8902 cells, DeltaTag (**6a**) exhibited a steeper dose-response curve than Deltazinone **1** with stronger growth inhibition at higher concentrations (91 ± 2% apparent growth inhibition by **6a** as compared to 62 ± 9% by Deltazinone **1** at 12.5 μM, Fig. 6c). Comparing to its reversible counterpart **6b**, DeltaTag (**6a**) was consistently more potent than **6b**, again with the exception in NCI-H358 cells for which both compounds exhibited similar activity (IC$_{50}$ of **6a** = 6.3 ± 3.5 μM; IC$_{50}$ of **6b** = 6.8 ± 1.9 μM, Fig. 6b). The consistent improvement in cellular potency from the reversible inhibitor **6b** to the covalent PDEδ inhibitor **6a** supported the design strategy for covalent inhibitor development. While Deltazinone **1** showed little or no growth inhibition for doses up to 50 μM in MIA PaCa-2-, SW480- and BxPC-3 cell lines, DeltaTag (**6a**) consistently exhibited stronger growth inhibitory effects in these cell lines (with IC$_{50}$ values of 7.0 ± 0.4 μM, 18.6 ± 3.0 μM and 20.0 ± 1.3 μM, respectively, Fig. 6b). In-cell target engagement, on the other hand, was supported by the observation that growth impairment of HAP1 PDEδ knockout cells by DeltaTag was less pronounced compared to HAP1 wild type cells at concentrations of 0 - 50 μM (paired $t$-test, two-tailed $p$-value = 0.049, Supplementary Fig. S5a). Potential off-target effects such as DNA alkylation were not observed with DeltaTag treatment at 10 μM in a Comet assay (Supplementary Fig. S5b). Furthermore, the compounds exhibited no general cytotoxicity at concentrations up to 25 μM, as demonstrated by cell viability and morphological profiling in a panel of human cell lines (Supplementary Fig. S5c, d). At non-toxic doses, the most significant improvement in cellular potency brought by DeltaTag (**6a**) was observed in KRAS-dependent MIA PaCa-2 cells (Fig. 6c, d).

## Discussion

We have developed and presented proof-of-principle for a covalent inhibition strategy aimed at the development of inhibitors targeting glutamic acids in protein binding sites. Employing the lipoprotein binding chaperone PDEδ and the previously developed reversible inhibitor Deltazinone as an example, we demonstrate the design and investigation of a warhead inspired by the HaloTag system. Structure-based design led to the discovery of DeltaTag with an alkyl bromide warhead, which covalently labels a glutamate residue in the prenyl binding pocket of PDEδ with sufficient reactivity. Given the electrophilic nature of alkyl bromides and potential risks of non-specific labelling at more reactive Cys and Lys side chains, we further investigated in vitro stability and labelling selectivity of DeltaTag. Gratifyingly, DeltaTag exhibits moderate stability, with low reactivity towards the most reactive sulphur nucleophile glutathione ($t_{1/2}$ for GSH adduct formation = 1035 h). In vitro labelling experiment with PDEδ further demonstrated that the labelling was selectively achieved at its binding site p.E88, as confirmed by mass spectrometric evaluation of the digested covalent adduct, with a parallel search for modifications at all nucleophilic residues, including the more reactive Cys and Lys. In comparison to haloacetamides, particularly more recently developed chlorofluoroacetamides (CFAs) and dichloroacetamides (DCAs) with mild reactivity towards cysteines ($t_{1/2}$ for GSH adduct formation in the range of 1000 min) and consistently low, tuneable cytotoxicity[7,41,42], DeltaTag shows much weaker reactivity, suggesting at least comparably low toxicity from non-specific labelling reactivity. CFAs and DCAs also offer tuneable labelling selectivity and reversibility of covalency under aqueous conditions, which help eliminate or at least reduce off-target labelling of solvent-exposed residues[42]. Similarly, alkyl bromide warhead targeting carboxylate residues may also hold promise for the development of targeted covalent ligands for its reduced off-target engagement with solvent-exposed residues, as the reversibility of ester bond formation could be significantly affected by protein micro-environment. In the complex proteome under physiological conditions, sustained covalent modification of PDEδ was also achieved by DeltaTag despite the counteracting release of ligands from the binding site by Arl2. Phenotypically, DeltaTag inhibits the proliferation of human cancer cell lines with an on-target activity supported by TPP and PDEδ knock-out cell model, and impacts the PDEδ-Rheb-mTOR pathway, at effective concentrations for which no off-target effects such as DNA alkylation or general cytotoxicity were observed.

In the context of rational TCI drug development, the 2-bromoethylbenzene side chain, as in DeltaTag, is also present as a fragment-sized soft electrophile in other covalent inhibitors such as PBRM, which demonstrated metabolic stability, favourable pharmacokinetic parameters and tolerance to high doses in mice, thereby supporting the general applicability of the phenethyl bromide warhead in bioactive compounds and medicinal chemistry programs[43]. DeltaTag, however, possesses physicochemical properties at the borderline of druglikeness and favourability for oral availability (Supplementary Table 3, predicted by SwissADME)[44]. While being stable in mouse plasma with no reactivity observed within an hour

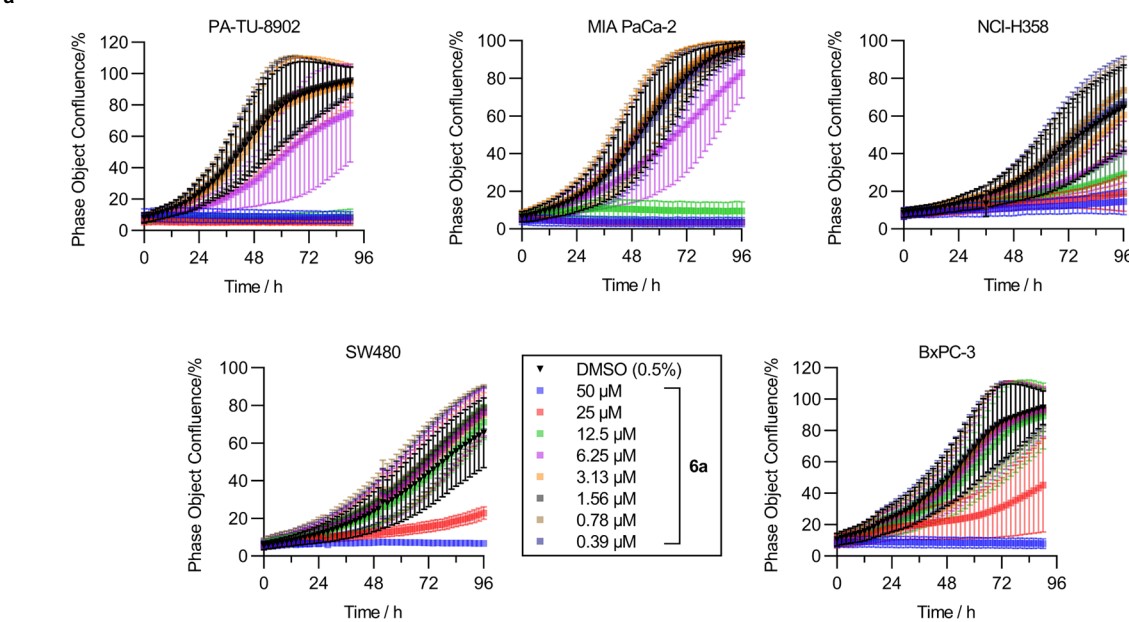

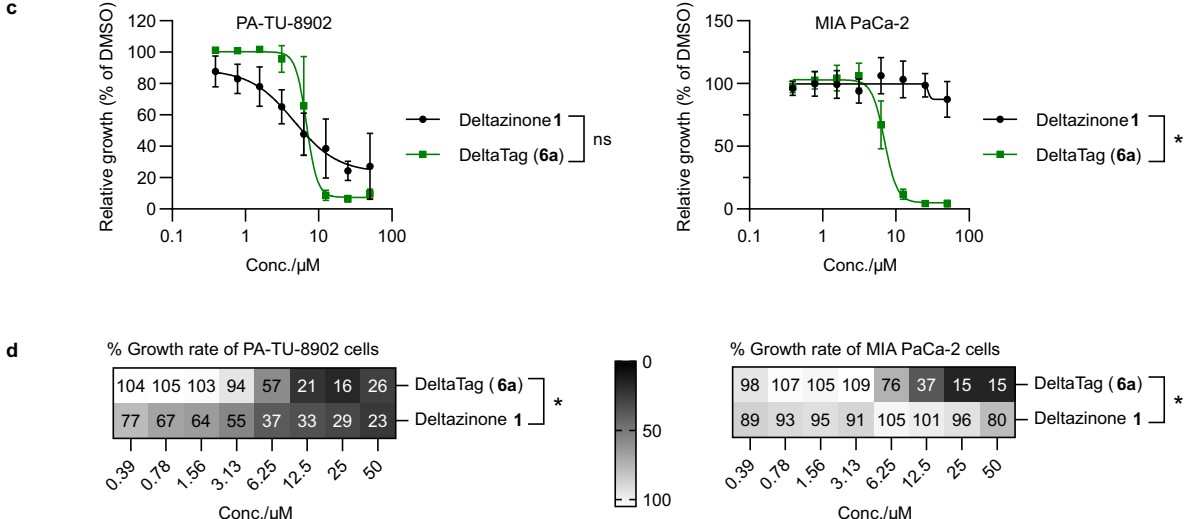

| Cell line | Tissue | KRAS mutation | Oncogenic KRAS dependency | IC$_{50}$ / µM | | |
|---|---|---|---|---|---|---|
| | | | | Deltazinone 1 | DeltaTag (6a) | 6b |
| PA-TU-8902 | Pancreatic | G12V | Yes | 4.6 ± 1.4 | 6.8 ± 0.5 | 12.0 ± 1.0 |
| MIA PaCa-2 | Pancreatic | G12C | Yes | - | 7.0 ± 0.4 | 10.9 ± 1.9 |
| BxPC-3 | Pancreatic | WT | No | - | 20.0 ± 1.3 | > 26 |
| NCI-H358 | Lung | G12C | Yes | 0.14 ± 0.11 | 6.3 ± 3.5 | 6.3 ± 1.9 |
| SW480 | Colorectal | G12V | Yes | - | 18.6 ± 3.0 | > 50 |

**Fig. 6 | DeltaTag exerts antiproliferative effects in KRAS-mutant cancer cell lines. a** Cell proliferation with dose-dependent treatment of DeltaTag (**6a**) monitored with real-time live-cell imaging by Incucyte. Data of per cent (%) phase object confluence were plotted as mean ± s.d. (representative of biological replicates $n = 4$ for PA-TU-8902, BxPC-3, NCI-H358 & SW480, $n = 3$ for MIA PaCa-2). **b** Cellular IC$_{50}$ values of DeltaTag (**6a**), **6b** and Deltazinone **1**, with an overview of KRAS mutation and oncogenic KRAS-dependence in respective cell lines[74–76]. Data were presented as mean ± s.d. (representative of biological replicated $n = 4$ for PA-TU-8902, BxPC-3, NCI-H358 & SW480, $n = 3$ for MIA PaCa-2). **c** Relative growth of selected cell lines after 72 h of compound treatment, normalised to DMSO control.

Data were plotted as mean ± s.d. (representative of biological replicates $n = 4$ for PA-TU-8902, $n = 3$ for MIA PaCa-2). Paired $t$-test, one-tailed $p$-value = 0.285, ns in PA-TU-8902; one-tailed $p$-value = 0.03 in MIA PaCa-2. **d** Growth rate in the selected cell lines by treatment with DeltaTag (**6a**) in comparison to Deltazinone **1**. Growth rates were estimated by AUC integration over 60 h after compound treatment, monitored by real-time cell analysis of per cent phase area confluence, normalised to DMSO control. Mean values of growth rates were plotted for the heat maps. Paired $t$-test comparing DeltaTag (**6a**) treatment to Deltazinone **1** treatment, in PA-TU-8902 cells, one-tailed $p$-value = 0.03; in MIA PaCa-2 cells, one-tailed $p$-value = 0.07. Source data are provided as a Source Data file.

(Supplementary Fig. S6a, b), DeltaTag was almost completely metabolised within 50 min with an intrinsic clearance of 117 µl/min/mg when tested in the presence of mouse liver microsomes (Supplementary Fig. S6c, d). This intrinsic clearance of DeltaTag roughly translates to an in vivo clearance of ~ 4 l/h/kg using the simplified well-stirred model, which falls into the high clearance category (mouse liver blood flow 5.4–7.2 l/h/kg)[45]. Notably, DeltaTag was metabolically stable in the absence of co-factor NADPH (Supplementary Fig. S6c,d), which strongly suggests CYP metabolism being the major driver. Additionally, its liver microsomal stability shows a linear decline in the semi-log plot with a decent half-life ($T_{1/2} = 11.8$ min, Supplementary Fig. S6d), suggesting that there is no significant CYP saturation or inhibition induced by DeltaTag. The current pharmacokinetic profile of DeltaTag, while demonstrating potential for further optimisation, limits further in vivo validation of its biological efficacy.

This proof-of-concept study in PDEδ suggests the potential of the approach inspired by HaloTag Technology to expand the toolbox of carboxylate-targeting electrophiles, to target proteins with hydrophobic binding sites that lack accessible reactive Cys and Lys residues, but rather contain less nucleophilic carboxylic acids. For instance, a survey of lipid-interacting proteins, such as enzymes involved in prenylation, acylation (myristoylation or palmitoylation)[46] and lipoprotein chaperones analogous to PDEδ[47–49], for Glu and Asp residues in a hydrophobic environment (see Supplementary Fig. S7 for more examples) indicated that UNC119, a chaperone for *N*-myristoylated proteins[50], by analogy to PDEδ, contains a glutamic acid in its lipid binding site, surrounded by hydrophobic amino acids (Supplementary Fig. S8a). The UNC119A inhibitor squarunkin A occupies the lipid-binding pocket and forms a hydrogen bond with glutamic acid[49], suggesting the possibility of covalent linkage by attachment of a suitable haloalkane-based warhead. For orienting exploration of this notion, we employed a fragment-based approach based on squarunkin A structure and replaced one side of its squaramide substituents with a bromopropyl side chain. After incubation of the ligand with the homologous UNC119B protein, we gratifyingly observed covalent modification at the equivalent glutamic acid in its lipid binding site (Supplementary Fig. S8 and Supplementary Table 4). The structure of this fragment has not been further evolved to increase the binding affinity and selectivity for the binding pocket of UNC119B. It was, therefore, not surprising that the fragment-sized alkyl bromide also reacted with other solvent-exposed glutamic and aspartic acids in the protein (Supplementary Table 4). These findings, on the one hand, indicate that the strategy for covalent targeting of glutamic and aspartic acids may not be restricted to the proof-of-principle example PDEδ investigated here. On the other hand, they demonstrate that proper compound design, most advantageously guided by structures of protein-noncovalent inhibitor complexes, would be instrumental to arrive at potent and selective covalent small molecule inhibitors for subsequent chemical biology and medicinal chemistry research. Future successful examples will be necessary to establish this approach as a mature and broadly applicable methodology for covalent targeting of carboxylates.

## Methods

### Compound synthesis and analysis
Details related to compound synthesis and analysis can be found in Supplementary Information, section 3—Chemical synthesis.

### Protein purification
All proteins were expressed in the *Escherichia coli* strain Rosetta (BL21DE3). Competent cells were transformed with the respective pET plasmids[18] and inoculated on a cell plate with TB medium supplemented with 100 µg/ml ampicillin and 30 µg/ml chloramphenicol for overnight incubation at 37 °C. A single colony was used to inoculate TB medium supplemented with 100 µg/ml ampicillin and incubated at 37 °C with simultaneous shaking overnight. The next day, 5 L of TB medium was inoculated with 50 ml of the Rosetta suspension and incubated at 37 °C until OD ~ 1.0. Cells were induced at OD ~ 1.0 with 100 µM isopropyl β-D-1-thiogalactopyranoside (IPTG) and incubated at 20 °C for 7 h. Cells were harvested and lysed in lysis buffer (30 mM Tris-HCl, pH = 7.5, 150 mM NaCl and 1 mM β-mercaptoethanol, 1 mM PMSF, i.e. phenylmethylsulfonyl fluoride) with sonication on ice. Supernatant of histidine-tagged protein was collected by centrifugation at $13,000 \times g$ and 10 °C for 35 min and subsequently loaded onto a Ni-NTA column (QIAGEN) and eluted with elution buffer (30 mM Tris-HCl, pH = 7.5, 150 mM NaCl, 1 mM dithiothreitol (DTE) and 250 mM imidazole), followed by gel filtration on a Superdex G75 S26/60 column using elution buffer without imidazole. Protein purity was checked by SDS-PAGE.

Nucleotide exchange of freshly purified GDP bound Arl2 protein was achieved by overnight incubation at 4 °C with 20 U alkaline phosphatase (Roche Diagnostics GmbH, Germany, #11097075001, 20 U/µl) and 2-fold excess of the non-hydrolysable GTP analogue (GppNHp tetralithium salt, NU-401-10, Jena Bioscience) in exchange buffer (30 mM Tris-HCl, pH = 7.5 with 150 mM NaCl, 0.1 mM ZnCl₂, 200 mM ammonium sulphate, 3% glycerol and 1 mM DTE). The next day, supernatant was purified with a HiTrap desalting column (GE Healthcare, GE17-1408-01) with elution buffer (30 mM Tris-HCl, pH 7.5, with 150 mM NaCl, 5 mM MgCl₂, 3% glycerol and 1 mM DTE). The exchange efficiency was checked by HPLC with a C-18 reversed-phase column with isocratic elution buffer (50 mM potassium phosphate, pH = 6.6, 10 mM tetrabutylammonium bromide and 8% acetonitrile) and detection wavelength at 252 nm, confirming exchange to Arl2 GppNHp by ≥95%.

### MALDI mass spectrometry
PDEδ (20 µM) was incubated with compounds (60 µM) in HEPES buffer (20 mM HEPES, 150 mM NaCl, pH = 7.5) with 0.6% DMSO at 37 °C for the specified time (2 h, 4 h, 6 h, 8 h, 24 h and subsequently every 24 h up to 7 days). The solution was briefly centrifuged before a sample was taken at the designated time for analysis by MALDI. A saturated solution of sinapinic acid (SA) in EtOH was used as matrix A and added to an MTP 384 ground steel target plate (Bruker) and dried in air. A saturated solution of SA in 30/70 acetonitrile (ACN)/H₂O with 0.1% TFA was used as matrix B. 1 µl of the sample was mixed with 2 µl of matrix B, and 1 µl of this mixture was placed on top of matrix A on the MTP 384 ground steel target plate and dried in air. Mass spectra were obtained over the m/z range 15,000–25,000 using a Bruker UltrafleXtreme XIAL DI-TOF/TOF mass spectrometer. Percentages of covalent adduct formation were estimated by the relative intensity of the respective peaks in MALDI spectra.

### Differential scanning fluorimetry (NanoDSF)
PDEδ (20 µM) was incubated with compounds (60 µM) or DMSO in HEPES buffer (20 mM HEPES, 150 mM NaCl, pH = 7.5) with 0.6% DMSO adjusted for all conditions at 37 °C for the specified time (2 h initially, and 24 h or 48 h to achieve near complete covalent modification indicated by MALDI). Afterwards, a sample of the solution was cooled down to room temperature and was loaded onto a capillary, followed by the measurement of fluorescence intensities at 350 nm and 330 nm from 20 °C up to 90 °C (with a rate of increment of 1 °C/min) using the Prometheus Panta device (NanoTemper® Technologies, DE). Melting curves were visualised by GraphPad Prism 9.5.1 (GraphPad software, USA) by plotting the first derivative of F 350 nm/F 330 nm over temperature, and the inflexion point is taken as the respective melting temperature.

### Fluorescence polarisation assay
The binding affinity of the fluorescence labelled analogue of the HMG-CoA reductase inhibitor atorvastatin (FA probe), which has previously

been shown to also bind to PDEδ[51], was validated and experimentally determined by a direct displacement assay to be $10.6 \pm 2.3$ nM (reported $K_D$ of FA probe[21] = $7.1 \pm 4$ nM, Supplementary Fig. S3). For a competitive fluorescence polarisation assay, the respective compound was 2× serially diluted in PBS buffer (containing 0.05% Chaps) in a black, non-binding round-bottom 384-well plate (Corning #4514) (10 uL/well) to a concentration range of 0–1000 nM, with 1% DMSO adjusted for all. To this solution, an equal volume of premixed solution of PDEδ (80 nM) and FA probe (48 nM) in the same PBS buffer (containing 0.05% Chaps, 1% DMSO) was added so that the final mixture contained compound concentration in the range of 0–500 nM, 40 nM PDEδ and 24 nM FA probe. The assay was further adapted to be performed at room temperature or at 37 °C and in the presence or absence of Arl2•GppNHp, such that its concentration in the solution was equivalent to PDEδ (40 nM). The plate was then sealed, briefly centrifuged and shaken overnight at room temperature at 600 rpm. After brief centrifugation on the next day, fluorescence polarisation values were measured by a plate reader (Tecan SPARK) with excitation wavelength at 485 nm, emission wavelength at 535 nm and temperature setting at 25 °C. $IC_{50}$ values were calculated by fitting anisotropy readouts against compound concentration with GraphPad Prism 9.5.1 (GraphPad software, USA) using a four-parameter variable slope non-linear regression curve fit.

## Compound stability in aqueous buffers and in the presence of GSH

Compound (1 mM) was incubated with the respective buffers (HEPES 20 mM, 150 mM NaCl, pH = 7.5; 0.2 M sodium acetate buffer, pH = 5.6; 0.2 M sodium citrate buffer, pH = 6.2; 0.2 M potassium phosphate buffer, pH = 7.4, 1 mM EDTA, with or without 10 mM GSH), with 1% DMSO adjusted for all conditions. The resulting solutions were incubated with shaking (600 rpm) at 37 °C, protected from light. At the respective time points (0, 2 h, 4 h, 6 h, 8 h, 24 h, 30 h, 48 h, 56 h, up to 72 h), a sample was taken and diluted in acetonitrile and analysed by HPLC-MS (Agilent Technologies 1290 Infinity, 6150 Quadrupole LC/MS). Per cent (%) compound remaining in the solution was estimated by the ratio of areas under the curve, and data plotted were fitted to pseudo-1st kinetics in the calculation of rate constants ($k$) and half-lives ($t_{1/2}$), where k is the negative slope of a linear regression of ln (% compound remaining) over time. Half-lives ($t_{1/2}$) in respective buffers and half-lives for GSH adduct formation ($t_{1/2}$ GSH) were calculated with the following equations:

$$t_{1/2} = \ln 2 / k \tag{1}$$

$$\begin{aligned} k(GSH) = &\ k(\text{total consumption in phosphate buffer in presence of GSH}) \\ &- k(\text{phosphate buffer in absence of GSH}) \end{aligned} \tag{2}$$

$$t_{1/2} GSH = \ln 2 / k(GSH) \tag{3}$$

## Mass spectrometry analysis of PDEδ adducts after Glu-C digestion

For each replicate, 1.5 µg covalent adduct of PDEδ sample in PBS buffer was denatured, reduced, alkylated and digested by Glu-C and desalted, together with a negative control of unbound vehicle-treated PDEδ sample prior to mass spectrometry analysis. In the denaturing step, each 4 µl sample was mixed with 18 µl of an 8 M guanidine hydrochloride (cas 50-01-1, Carl Roth, #0037.1) solution. The final mixture containing 6.5 M guanidine hydrochloride was heated at 95 °C for 15 min. Subsequently, after cooling, 0.5 µl of a 50 mM dithiothreitol (DTT, cas 3483-12-3, Gerbu Biotechnik, #1008-100 g) solution was added. Samples were reduced with a final concentration of 1 mM DTT,

with incubation at 60 °C for 20 min. 2.5 µl of a 50 mM 2-chloroacetamide (CAA, cas 79-07-2, Sigma Aldrich, #22790) solution was then introduced, achieving a final concentration of 5 mM CAA. Samples were incubated in the dark at room temperature for 30 min. Afterwards, guanidine hydrochloride concentration was reduced below 0.8 M by adding 225 µl of a 20 mM ammonium bicarbonate (cas 1066-33-7, Sigma Aldrich, #A6141-500g) solution to each sample. For enzymatic digestion by Glu-C, 1.5 µl of Glu-C (reconstituted as 0.05 µg/µl, Promega V1651) was added, maintaining an enzyme-to-protein ratio of 1:20 ($w/w$). Samples were incubated at 37 °C overnight with shaking at 400 rpm, protected from light. 5 µl of 10% trifluoroacetic acid (TFA, cas 76-05-1, Sigma Aldrich, #302031) was added to each sample to quench the reaction. Desalting was carried out with stage tips with C18 extraction discs (Empore™ high performance extraction discs, 47 mm, 3 M Bioanalytical Technologies #2215). Each stage tip, which consists of two stacked layers of C18 discs, was activated with methanol (100 µl), washed once with buffer B (100 µl, 0.1% formic acid, 80% acetonitrile), and twice with buffer A (100 µl/time, 0.1% formic acid). After sample loading and washing with buffer A once, the desalted sample was eluted with buffer B (20 µl/sample) and centrifuged at 500×g at room temperature for 5 min. Eluted samples were then dried in a SpeedVac at 30 °C.

For nanoHPLC-MS/MS analysis, samples were redissolved in 15–20 µl of 0.1% TFA and 3 µl each were loaded onto a pre-column cartridge for desalting. Desalting was performed for 5 min using 0.1% TFA as eluent with a flow to waste (30 µl/min) followed by back-flushing of the sample during the whole analysis from the pre-column to the PepMap100 RSLC C18 nano-HPLC column (2 µm, 100 Å, 75 µm ID × 50 cm, nanoViper, Dionex, Germany) using a linear gradient starting with 95% solvent A ($H_2O$ with 0.1% formic acid)/5% solvent B (acetonitrile with 0.1% formic acid) and increasing to 30% solvent B after 35 min (measurements on Q-Exactive Plus) or increasing stepwise to 20% solvent B after 37 min and further to 32% solvent B after 44 min (measurements on Q-Exactive HF) using a flow rate of 300 nl/min. The nano-HPLC was online coupled to a Quadrupole-Orbitrap Q-Exactive Plus or a Q-Exactive HF Mass Spectrometer using an uncoated SilicaTip (ID 20 µm, Tip-ID 10 µM). Mass range of $m/z$ 300 to 1650 (Q-Exactive Plus) or 375 to 1500 (Q-Exactive HF) was acquired with a resolution of 70000 (Q-Exactive Plus) or 120,000 (Q-Exactive HF) for full scan, followed by up to ten (Q-Exactive Plus) or fifteen (Q-Exactive HF) high energy collision dissociation (HCD) MS/MS scans of the most intense at least doubly charged ions using a resolution of 17500 (Q-Exactive Plus) or 15000 (Q-Exactive HF) and a NCE energy of 25% (Q-Exactive Plus) or 27% (QExactive HF).

Data evaluation was performed using MaxQuant software (v.2.2.0.0)[52]. The spectra were queried against the PDE6D sequence (Uniprot ID: O43924) and a contamination database with a 1% false discovery rate, utilising a decoy database to analyse the false discovery rate. For database search, oxidation of methionine and N-terminal acetylation, carbamidomethylation of cysteines, and artificial modification of amino acids cysteines (C), aspartic acids (D), glutamic acids (E), threonine (T), serine (S), tyrosines (Y), lysine K, or histidines (H) have been set as variable modifications. Cleavages by Glu-C were analysed based on the enzyme that was used for digestion. All experiments were performed in three biological triplicates.

## Mass spectrometry analysis of UNC119B adducts after Lys-C digestion

Samples of UNC119B (200 µg protein per sample), either treated with compound 7 or vehicle DMSO, were processed by denaturation, reduction, alkylation, and digestion with rLys-C (Promega V1671), followed by desalting prior to mass spectrometry analysis. The experiment was conducted with three biological replicates. Mass spectrometry experiments were carried out using an Orbitrap Fusion LUMOS instrument (Thermo) paired with a Vanquish Neo ultra-

performance liquid chromatography (UPLC) system (Thermo). The UPLC system was configured in one-column mode, with the analytical column consisting of a fused silica capillary (75 μm × 28 cm) equipped with an integrated fritted emitter (CoAnn Technologies), packed in-house with 1.7 μm Kinetex core-shell beads (Phenomenex). The analytical column was housed within a column oven (Sonation PRSO-V2), which was maintained at 50 °C throughout both sample loading and data acquisition, and connected to a nanospray flex ion source (Thermo). The LC system utilised two mobile phases: solvent A (0.2% FA, 2% ACN, 97.8% H₂O) and solvent B (0.2% FA, 80% ACN, 19.8% H₂O), all solvents of UPLC grade from Honeywell. Peptide samples were loaded directly onto the analytical column at a maximum flow rate, typically ranging between 0.4 and 0.6 μl/min, so that the pressure would not surpass the set limit of 980 bar. Following loading, samples were separated on the analytical column using a 60- or 67-min gradient of solvent A and B (refer to the LC settings table for specifics) at a flow rate of 250 nl/min. The mass spectrometer operates with an Orbitrap Fusion Lumos Tune Application (version 4.1.4244) and Xcalibur software (version 4.7.69.37). The modified peptides were analysed at two different concentrations (5000 ng or 4000 ng/2000 ng for the first replicate) and using two distinct separation methods (60-min or 67-min solvent gradients, respectively). Further details regarding MS and LC settings are provided in the Supplementary Information–"Methods" section.

RAW spectra were submitted to a closed MSFragger (version 4.1)[53] search in Fragpipe (version 22)[54] using the "LFQ-MBR" workflow (label-free quantification and match-between-runs; default settings were used unless otherwise stated). RAW files were listed in the "Input LC-MS Files" section and experiment set "by file name". As "Data Type", we kept the default "DDA" (data-dependent acquisition). The MS/MS spectra were searched against a custom database generated in Fragpipe (2025-08-20-decoys-contam-UP000000625_83333_plus_-SOI_v01.fasta.fas (9040 entries)) containing the Uniprot *E. coli* reference proteome (UP000000625_83333.fasta; 4402 entries), the sequence of interest (UNC119B-6xHis), contaminants and decoys (the last two appended and generated by Fragpipe in the Database section). MSFragger searches allowed oxidation of methionine residues (16 Da; 3) and acetylation of the protein N-terminus (42 Da; 1), Carbamylation at Lysin (43 Da; 3) and peptide N-terminus (43 Da; 1) and modification of glutamic acids and aspartic acids by compound **7** (307 Da; 2) as variable modification (first value in brackets refers to the molecular weight of the modification, second value to the maximum number of occurrences per peptide). A maximum of three variable modifications and a maximum of five combinations were allowed globally. Carbamidomethylation on Cysteine (57 Da) was selected as a static modification. Enzyme specificity was set to "LysC-p SEMI". The initial precursor and fragment mass tolerance was kept at ±20 ppm. Mass calibration and parameter optimization was selected. Validation of peptide spectrum matches was done using MSBooster[55] using DIA-NN[56] for RT and spectra prediction. Peptide spectra matches (PSM) were validated using Percolator with a minimum probability setting of 0.5. Protein inference was performed using ProteinProphet (part of Philosopher version 5.1.1)[57]. The final reported protein FDR was 0.01 (based on the target-decoy approach). Protein quantification was performed with IonQuant (version 1.10.27). Add MaxLFQ (min ions 1), MBR (FDR 0.01) and normalisation of intensity across runs was selected. Unique and razor peptides were allowed. Advanced options were kept at the default. Further analysis and filtering of the results were done in Perseus v1.6.10.0[58]. Comparison of protein group quantities (relative quantification) between different MS runs is based solely on the LFQs as calculated by IonQuant, the MaxLFQ algorithm.

## X-ray crystallography

To obtain crystal structures of covalent adducts, compounds **5e** and **6a** were incubated overnight at 37 °C with PDEδ at a 10:1 molar ratio (2 mM compound: 200 μM protein) in HEPES buffer (20 mM HEPES, 150 mM NaCl, pH = 7.5) containing 1% DMSO. The resulting mixtures were then washed and concentrated to approximately 20 g/l using Amicon 3 kDa molecular weight cutoff filters (Millipore, UFC5003BK) in protein buffer containing 30 mM Tris-HCl, 150 mM NaCl, 1 mM *β*-mercaptoethanol, pH = 7.5. Crystallisation trials employed the sitting-drop vapour diffusion method, mixing 0.1 μl of protein solution with 0.1 μl of precipitant solution in MRC 3-drop plates (Jena Bioscience, UK). Crystals suitable for diffraction grew within 12 days at 20 °C using a precipitant solution containing 4 M sodium formate (Qiagen Classics suite). These crystals were harvested, cryoprotected by supplementing the mother liquor with 25% glycerol, and flash-frozen in liquid nitrogen. Diffraction data for the PDEδ•**5e** complex were collected at the Swiss Light Source (SLS) beamline X10SA, while data for PDEδ•**6a** were collected on a Bruker D8 Discover home source equipped with a Bruker Photon III detector and Diffrac software. Both datasets were processed and scaled by XDS and XSCALE[59]. For solving the crystal structures by PHASER[60] within the PHENIX software suite[61], the co-crystal structure of PDEδ•Deltazinone derivative with the PDB accession code 5E80 (chain A) was used as the model for molecular replacement. Subsequent refinement involved iterative cycles using COOT[62] for manual model building and phenix.refine[63]. Ligand topologies were generated from SMILES strings using AceDRG[64] within the CCP4 suite[65], with covalent bond geometry restraints manually adjusted in the phenix.refine parameter file. Protein-ligand interactions were analysed using LigPlot+[66] and Pymol (Version 2.5.4, Schrödinger, LLC). The final structures were deposited in the Protein Data Bank under accession codes 9HMC (PDEδ•**5e**) and 9HMD (PDEδ•**6a**), with detailed data collection and refinement statistics provided in Supplementary Information–Supplementary Table 5.

## Cell lines and culture conditions

All mammalian cells were cultured and maintained in a sterile environment with a humidified atmosphere at 37 °C and 5% CO₂. Jurkat (ACC282, RRID: CVCL_0065), PA-TU-8902 (ACC179, RRID: CVCL_1845), BxPC3 cells (ACC760, RRID: CVCL_0186), HEP-G2 (ACC180, RRID: CVCL_0027) were purchased from DSMZ GmbH (Germany). MIA PaCa-2 (CRM-CRL-1420, RRID: CVCL_0428), SW480 (CCL-228, RRID: CVCL_0546), HEK293T (ATCC-CRL-3216, RRID: CVCL_0063) and CaCo-2 (ATCC-HTB-37, RRID: CVCL_0025) cells were obtained from ATCC (USA), and NCI-H358 (ATCC-CRL-5807, RRID: CVCL_1559) from LGC Standards (Germany). U2OS (CLS-300364, RRID: CVCL_0042) cells were obtained from CLS Cell Lines Service GmbH (Germany). HAP1 wild type (Horizon #C631, RRID: CVCL_Y019) and HAP1 PDE6D knockout (Horizon #HZGHC006484c003, RRID: CVCL_XR47) cells were purchased from Horizon (Horizon Discovery, UK). Jurkat, BxPC3, NCI-H358, SW480 and HEP-G2 cells were cultured in RMPI-1640 medium (P04-18047, PAN Biotech) with 10% of Foetal Bovine Serum (Gibco, #10270-106) and 1% non-essential amino acids (NEAA, P08-32100, PAN Biotech). PA-TU-8902, MIA PaCa-2, HEK293T and U2OS cells were cultured in Dulbecco's Modified Eagle's medium (DMEM, P04-03550, PAN Biotech) supplemented with 10% FBS, 1% NEAA and 1 mM sodium pyruvate (P04-43100, PAN Biotech). CaCo-2 cells were cultured in Eagle's Minimum Essential Medium (EMEM, P04-08056, PAN Biotech) supplemented with 10% FBS. HAP1 wild-type and PDE6D knockout cells were cultured in Iscove's Modified Dulbecco's Medium (IMDM, P04-20350, PAN Biotech) supplemented with 10% FBS. Mycoplasma tests carried out on a regular basis confirmed that cells were free of contamination at all times.

## In-lysate CETSA and TPP

Jurkat cells were grown to a density of 1.0–2.5 × 10⁶ cells/ml. A 100 ml aliquot of the suspension was divided equally into two Falcon tubes and chilled on ice for 2 min. Cells were pelleted by centrifugation at 350×*g* for 3 min at room temperature. After discarding the

supernatant, the combined pellet was resuspended in 25 ml of ice-cold PBS buffer. The washing step—centrifugation at 350×$g$ for 2 min at room temperature, followed by supernatant removal—was further repeated twice. Following the final wash, the cell pellet was resuspended in 1.5 ml of PBS containing 0.4% ($v/v$) NP40 alternative and flash-frozen in liquid nitrogen. Lysis was achieved through five to six iterative freeze-thaw cycles: frozen samples were thawed at 23 °C in a thermomixer (Thermomixer Comfort, Eppendorf) until 60-80% liquefied, then held on ice until fully thawed, before being rapidly refrozen in liquid nitrogen. The resulting lysate was separated by ultracentrifugation at 100,000×$g$ for 20 min at 4 °C (Beckman Coulter Optima MAX-XP, MLA-80 rotor). The supernatant was carefully collected, snap-frozen in liquid nitrogen, and stored at −80 °C for subsequent analysis.

Protein concentration was quantified using the Bradford protein assay. Samples were diluted with PBS to a final concentration of 2.5 mg/ml and split into two aliquots of equal volume. One aliquot was treated with 10 µM compound **6a** (achieved by adding 1.7 µl of a 10 mM DMSO stock to 1.7 ml sample, yielding 0.1% DMSO final concentration), while the control aliquot received an equivalent volume of DMSO alone. Both aliquots were incubated at 37 °C for either 15 min or 2 h. Following incubation, each aliquot was further subdivided into nine 120 µl portions in PCR tubes. These portions underwent thermal denaturation for 5 min at precisely defined temperatures (37.0, 41.7, 45.8, 49.7, 53.5, 57.4, 61.3, 64.3, 67.0 °C) using a MasterCycler EpGradient S thermal cycler (Eppendorf SE, DE). Samples were then cooled to room temperature and centrifuged at 100,000×$g$ for 20 min at 4 °C (Beckman Coulter Optima MAX-XP, TLA-120.1 rotor). The supernatant from each sample was carefully collected into low protein-binding tubes, snap-frozen in liquid nitrogen and kept at −80 °C for storage. For analysis, 75 µl of each supernatant was allocated for mass spectrometry-based thermal protein profiling (TPP), while the remaining 25 µl was used for immunoblotting to assess PDEδ levels. Western blot band intensities were quantified using Image Lab software and normalised to the intensity of the first band thermal denatured at the 45.8 °C condition.

For mass spectrometric analysis, sample processing was further proceeded with reduction and alkylation, followed by acetone precipitation and tryptic digestion. Each digested aliquot was then labelled with isobaric TMT reagents (Thermo Fisher Scientific, A52047) corresponding to its specific temperature condition from the thermal shift assay. Detailed protocols for sample handling, nanoHPLC-MS/MS instrumentation parameters, and data processing methodologies are provided in the Supplementary Information—"Methods" section.

## CETSA in intact cells (In-cell CETSA)

Jurkat cells (adjusted to 2 × 10⁶ cells/ml, 5 ml/flask) were seeded in two T25 tissue culture flasks and treated with 10 µM compound or vehicle (DMSO), with 0.1% DMSO adjusted, for 15 min at 37 °C. Cells were collected and resuspended in cold PBS and further washed thrice in cold PBS. The sample from each treatment condition was distributed equally into twelve tubes and subjected to heating at different temperatures (the first 9 tubes with one gradient cycle with temperatures 37.0, 41.7, 45.8, 49.7, 53.5, 57.4, 61.3, 64.3, 67.0 °C and the remaining 3 tubes with a second gradient cycle with temperatures 70.0, 73.6, 82.7 °C to completely precipitate PDEδ) in the MasterCycler EpGradient S (Eppendorf SE, DE). Afterwards, NP40 alternative was added to a final concentration of 0.4% ($v/v$), and cells were lysed by five consecutive freeze/thaw cycles. Soluble fractions were separated from denatured proteins by ultracentrifugation at 100,000×$g$ and 4 °C for 20 min (Beckman Coulter Optima MAX-XP, with TLA-120.1 rotor). Supernatants were transferred to new tubes, and equal volumes of each sample were loaded on an SDS-PAGE gel and subjected to immunoblot analysis. Obtained band intensities were quantified using

Image Lab software and normalised to the intensities of the bands at 37 °C.

## FLIM

HEK293T cells were transiently co-transfected to overexpress mCitrine-Rheb and mCherry-PDEδ (1:1.2 ratio of cDNA), with the plasmids previously described[22]. Fluorescence lifetime images were acquired using a confocal laser-scanning microscope (Leica SP8). For the detection of the donor mCitrine, the sample was excited with a supercontinuum White Light Laser (WLL) with a notch line filter at 470 nm at a 40-MHz repetition frequency. Fluorescence signals were collected through an oil immersion objective and spectrally filtered using a narrow-band emission filter and detected with a photon-counting HyD detector from 519 to 541 nm. Images were analysed in real-time on a global scale using the built-in Leica Application Suite X (LAS X) for FLIM, with lifetime calculated with fitting to the in-built model of mono-exponential reconvolution.

## Phosphoproteomics

NCI-H358 cells (5 × 10⁶ cells/dish) were seeded in two 10 cm dishes and incubated in a humidified atmosphere at 37 °C and 5% $CO_2$ overnight. Cells were then treated with 10 µM of compound **6a** or DMSO, with 0.1% DMSO adjusted for both conditions in fresh medium for 8 h incubated in a humidified atmosphere at 37 °C and 5% $CO_2$. All details related to subsequent sample preparation, nanoHPLC-MS/MS analysis and data evaluation can be found in the Supplementary Information —"Methods" section.

## Immunoblotting and immunofluorescence staining

Gel electrophoresis: unless otherwise noted, SDS-PAGE was run with self-made 12% Tris-glycine SDS-polyacrylamide gel in Tris-glycine SDS running buffer (25 mM Tris, 0.2 M glycine, 0.1% SDS) at 90 V for 15 min stacking and followed by 120 V until sufficient separation was visualised by PageRuler™ pre-stained protein ladder. For visualisation of covalent PDEδ adducts by western blot, SDS-PAGE was run with mPAGE™ Bis-Tris precast gels (4–12% or 4–20% acrylamide) in mPAGE™ MES running buffer at 125 V constant until maximum band resolution. Proteins were transferred onto a polyvinylidene difluoride (PVDF) membrane (Thermo Fisher Scientific, #88518, 0.45 µm) using a wet-tank blotting system (Bio-Rad) with pre-cooled transfer buffer (25 mM Tris, 0.2 M glycine, 10% methanol) at 100 V for 30 min.

For detection of vinculin as loading control, membranes were blocked with Intercept Blocking Buffer (LI-COR Biosciences, #927-70001) for 1 h at room temperature, incubated with the primary antibody anti-vinculin (Sigma-Aldrich, V9131, RRID: AB_477629, 1:5000) overnight at 4 °C in blocking buffer, then washed with PBS-T (PBS with 0.1% Tween-20) five times and incubated with IRDye 800CW-conjugated secondary antibody (LI-COR Biosciences, #926-32210, RRID: AB_621842, 1:5000) in blocking buffer for 1 h at room temperature with protection from light. For detection of PDEδ, loading control β-actin, as well as phosphorylated S6 ribosomal protein (2F9) (pS6P on S235/S236) and total S6 ribosomal protein (54D2) (tS6P), membranes were blocked with 5% milk in PBS-T for 1 h at room temperature, followed by washing, incubation with the primary antibody, anti-PDEδ (Invitrogen, PA5-22008, RRID: AB_11154288, 1:500), or anti-β-actin (Abcam, ab8227, RRID:AB_2305186, 1: 5000), or anti-pS6P (Cell Signalling Technology, #4856, RRID: AB_2181037, 1:1000), or anti-tS6P (Cell Signalling Technology, #2317, RRID: AB_2238583, 1:500) overnight at 4 °C and subsequently with horseradish peroxidase (HRP)-conjugated secondary antibodies (Invitrogen, #62-6520, RRID: AB_88369 for anti-mouse, 1:5000; Invitrogen, #31460, RRID: AB_228341 for anti-rabbit, 1:5000) for 1 h at room temperature. After washing with PBS-T five times, secondary antibody-incubated membranes were imaged with the ChemiDocMP Imaging System (BIO-RAD

Laboratories) directly with the IRDye800CW channel for visualisation of vinculin bands, or with the addition of western blotting detection reagent (SuperSignal™ West Dura Extended Duration Substrate, Thermo Fisher Scientific, #34075) for the HRP conjugated system with chemiluminescence channel. Relative band intensities were quantified using Image Lab (BIO-RAD).

Immunostaining: compound-treated PA-TU-8902 cells were washed once with cold PBS, fixed with 3.7% paraformaldehyde in PBS by incubation at room temperature for 10 min and permeabilised with 0.3% Triton X-100 in PBS by incubation at room temperature for 15 min. Cells were subsequently blocked by 2% BSA in PBS-T (0.1% Tween 20 in PBS) for 1 h at room temperature, incubated with the primary antibody Anti-Rheb (Santa Cruz Biotechnology Cat# sc-271509, RRID: AB_10659102, 1:500) overnight at 4 °C in blocking buffer, then washed with PBS-T three times and incubated with Alexa-555-conjugated secondary antibody (Invitrogen # A-31570, RRID:AB_2536180, 1: 1000) and DAPI (1 μg/ml) in blocking buffer for 1 h at room temperature with protection from light. After washing with PBS-T three times, PBS-incubated cells were imaged with Observer Z1 (Carl Zeiss, Germany) with a 63× oil objective.

### Real-time cell analysis for viability

Real-time cell analysis (RTCA) was conducted using an Incucyte S3 live-cell analysis system (Sartorius AG, Germany). Adherent cell lines were seeded at $5 \times 10^3$ cells per well in 96-well plates containing 100 μl of culture medium and incubated overnight at 37 °C with 5% $CO_2$. Following attachment overnight, the medium was replaced with fresh medium containing test compounds, with all conditions adjusted to contain 0.5% DMSO. Plates were then transferred to the Incucyte S3 instrument maintained at 37 °C and 5% $CO_2$. Phase-contrast images (2 images per well) were automatically captured at 2-h intervals until control wells (DMSO-treated) reached full confluency. Cell growth was quantified using the instrument's integrated Basic Analyser module by applying a confluence mask to calculate the percentage of phase area confluence. For dose-response assessment, confluence values at 72 h post-treatment were normalised to the DMSO control and analysed using GraphPad Prism 9.5.1 (GraphPad Software, USA). Dose-response curves were generated and fitted via four-parameter variable slope nonlinear regression to determine cellular $IC_{50}$ values. Growth rates were evaluated by plotting per cent confluence against time for each condition; the AUC for the first 60 h post-treatment was calculated in GraphPad Prism and normalised to the DMSO control.

### Reporting summary

Further information on research design is available in the Nature Portfolio Reporting Summary linked to this article.

## Data availability

All data generated or analysed during this study are available within the article and its Supplementary Information files, and from the corresponding author upon request. The proteomics data related to DeltaTag have been deposited in MassIVE with the following accession codes: MSV000096863, PXD059864 (https://doi.org/10.25345/C5GH9BN3J, artificial modification of PDEδ by **5e**) and MSV000096865, PXD059867 (https://doi.org/10.25345/C5736MD8P, artificial modification of PDEδ by **6a**), MSV000096867, PXD059870 (https://doi.org/10.25345/C5ZK55Z8F, phosphoproteomic profiling of **6a**), and MSV000096869, PXD059872 (https://doi.org/10.25345/C5Q23RC0W, TPP of **6a**). The mass spectrometry proteomics data for the UNC119 experiment have been deposited to the ProteomeXchange Consortium via the PRIDE partner repository with the dataset identifier PXD067823. The crystal structures of PDEδ modified by compounds **5e** and **6a** (DeltaTag) were deposited in the Protein Data Bank (PDB) with the accession numbers 9HMC and 9HMD respectively. Other crystal structures analysed in this study are available in the PDB database with the following accession numbers: 4JVF, 5E80, 5ML3, 5NAL, 6ZVY, 1HH4, 3PZ2, 6QGS, 5L7K and 7OK7. Source data are provided with this paper.

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

## Acknowledgements

Research at the Max Planck Institute of Molecular Physiology is supported by the Max Planck Society. We thank Professor Dr Daniel Rauh and Dr Matthias Müller for the possibility to use the Bruker D8 Discover home X-ray source at the Technical University of Dortmund (TU Dortmund) for the collection of crystallography data. We acknowledge the SLS for the provision of synchrotron radiation facilities, and we would like to thank the local beamline support scientists for assistance and support in using beamline X10SA. We thank Dr Belén Lucas and Dr María Lucas for assistance in crystal structure analysis. We are grateful to Dr Slava Ziegler for discussions of biological experiments. We thank Dr Sven Müller, Dr Michael Schulz and Dr Malte Schmick for support in FLIM microscopy and discussion of FLIM-FRET experiments. We thank Andreas Brockmeyer, Jenny Borman and Svenja Heimann for support in mass spectrometry measurements, Christine Nowak for protein and plasmid purification, Sasikala Thavam, Nathalie Bleimling, Jens Warmers and Anna Sophie Sickau for technical support. Parts of the graphical abstract were created in BioRender. Zhang, R. (2026) https://BioRender.com/n4ldba7.

## Author contributions

R.Z., J.L., and H.W. designed the project. J.L. conceptualised the design strategy with inspiration from the HaloTag technology and proposed compound structures. J.L. and R.Z. designed the compound library and performed organic synthesis of the compounds. J.L. performed initial biological screening, and R.Z. adapted the assays and performed all biological experiments. R.G. solved the crystal structures. R.Z. and R.G. analysed the crystallographic data. P.J., F.K. and M.K. supervised parts of the proteomics experiments. R.Z., F.K., J.L., and P.J. analysed the proteomics data. A.U. performed and analysed pharmacokinetic experiments. R.Z. wrote the original draft, and R.Z., J.L., and H.W. wrote and edited the manuscript. All authors discussed the results and commented on the manuscript.

## Funding

## Competing interests

The authors declare no competing interests.
