## [Transparent Peer Review file · Nature Communications]

Covalent Modification of a Glutamic Acid Inspired by HaloTag Technology

Corresponding Author: Professor Herbert Waldmann

Version 0:

Reviewer comments:

Reviewer #1

(Remarks to the Author)

The authors designed and synthesized series of covalent PDE δ inhibitors with alkyl bromide warheads using HaloTag Technology. Among them, the compound DeltaTag (6a) shows the best performance. It can overcome Arl2-mediated release, regulate signal transduction through the mTOR signaling pathway, and inhibit cancer cell proliferation. However, this study suffers from several significant shortcomings.

Most haloalkyl drugs, such as nitrogen mustard alkylating agents, exert their antitumor effects by targeting DNA, disrupting its structure and function, and ultimately preventing cell division and proliferation. These drugs are associated with substantial toxicity. In this study, the authors failed to address the potential off-target interaction between DeltaTag and DNA or evaluate the resulting toxic side effects. Furthermore, while the hypothesis that "reversible PDE δ inhibition is efficiently counterbalanced by Arl2/3-mediated inhibitor release calling for covalent inhibitor development" forms the foundation of this study, the improvement in the antitumor activity of the compound DeltaTag is extremely limited. Moreover, its observed antitumor activity may be directly attributable to its alkylating properties rather than specific PDE δ inhibition. Additionally, there are numerous errors with the compound structures and spectra data, raising concerns about the reliability of the findings. Therefore, this manuscript is not suitable for publication in NC.

Comments:

1. A major concern is whether PDE δ is a druggable target. The authors have developed numerous low nanomolar PDE δ inhibitors, which generally showed weak antitumor potency. The covalent PDE δ inhibitors designed herein failed to address these limitations, which largely undermined the significance of this work.
2. The authors should evaluate the interactions between the covalent PDE δ inhibitors with alkyl bromide warheads and DNA, and assess their toxicity in normal cells.
3. Figure 4. The authors should rule out the influence of temperature. Additional experimental group with compound 6a incubated at 37°C should be included.
4. Figure 4. The fluorescence polarization method is unsuitable for assessing Arl2-mediated release due to the competitive binding relationship between FA and PDE δ inhibitors. The observed decrease in anisotropy values with increasing concentrations of compound 6a directly reflects this competitive binding. Moreover, the lack of concentration dependence in the Arl2-treated groups further undermines the validity of this assay.
5. The authors should verify whether the compound DeltaTag can overcome the Arl2-induced release in cells.
6. Figure 5A. The interaction between small molecules and proteins may either enhance or reduce protein thermal stability. The authors exclusively considered increased stability while ignoring potential destabilization effects that could equally indicate DeltaTag interactions.
7. In Figure S4, the antitumor activity of the covalent inhibitor 6a is not improved compared to the reversible inhibitor 6b. Notably, both 6a and 6b, as haloalkyl compounds, exhibit significantly superior antitumor activity compared to the PDE δ inhibitor Deltazinone 1. This raises the possibility that their observed activity may be directly related to their alkylating properties rather than specific PDE δ inhibition.
8. The covalent inhibitors were not evaluated in animal models. More specifically, pharmacokinetic profiles, in vivo antitumor potency and in vivo toxicity remain to be assayed, which is important to confirm the potential advantages of covalent inhibitors.
9. In Figure 6, the authors have made theoretical predictions but have not provided direct experimental evidence to support the broader applicability of this approach.
10. There are numerous errors in the compound characterization data (Supplementary Materials). The details are listed in the review attachment.

Reviewer #2

(Remarks to the Author)
Publication with minor revisions

Zhang and co-workers have developed a covalent inhibitor for PDE, DeltaTag, based off inspiration from chloroalkane system. They utilize an alkyl bromide handle to selectively react with a glutamic acid deep in the active site of PDE, and displayed multiple downstream effects of this inhibition in live cells. Overall, the work is thorough and robust and a good contribution to Nature Communications, though the authors need to address the following concerns.

- The title of the paper, caption of abstract figure, and major portions of the introduction discuss bioconjugation techniques to Asp/Glu. This paper modifies only one Asp/Glu in PDE, and is not applicable to global Asp/Glu profiling, though the title and intro suggest this is a broadly applicable approach to modify any Asp/Glu residue. Please adjust language to clarify you are only targeting one specific Glu.
- All individual data points should be shown in all graphs including Fig 4, Fig 5g, Fig S3, and other applicable locations.
- It is unclear to me why 6b is reversible. While Br is a better leaving group (for 6a), I am surprised 0% of 6b (or 5h), especially since this paper is framed around the chloroalkane HaloTag which explicitly uses Cl as its leaving group. Furthermore, the IC₅₀ of 6b is ~1/2 of 6a, making this an even more interesting discovery. A discussion around this discrepancy should be added.
- Titles of Supplementary Figures should come before their respective graphs and writing, not in the middle between the figure and the writing.
- At the bottom of Page 8 the authors state "...which was reflected in the enhancement of thermostability from T_m = 13.5 °C to T_m = 15.8 °C..." without providing statistical context. Is this change significant? No p-values are provided in Fig 2e supporting this claim.
- Fig 2f: Why is the energy change for the T_m of PDE -DMSO controls almost negligible? The SI states all experiments were conducted with equal concentrations, yet the controls give ~10% of the signal from the PDE -6a samples?
- All protein intact MS figures should have the exact mass displayed about the peak. Notably in Fig 2c and Fig S3a.
- Fig S3a: The intensity units are drastically different between samples, ranging from upwards of 5000 a.u. to as low as 30 a.u. and seemingly decrease with added temperature. This is confusing as the SI states these experiments were done at the exact same concentrations. Please repeat with a similar injection volume for each data point.
- Fig 5g: The use of * is for general ranges of p-values, as the authors utilized in Fig 4, not for exact p-values. Additionally, without brackets, it is difficult to determine what comparison the p-value is referring to.
- Bottom of Page 16: The authors state "Overall, 6a,b exhibited a stronger growth inhibitory effect than Deltazine 1 in these cell lines with the exception of NCI-H358". This is false, as Fig S3 states that in cell line PA-TU-8902, 1 has lower IC₅₀ than either 6a or 6b. The authors should discuss why 1 is significantly better for PA-TU-8902 and NCI-H358, but had almost no effect on the other oncogenic KRas dependency lines (MIA-PaCa-2 and SW-480)
- Why is 6a effective at growth inhibition for non-oncogenic KRas dependent cell line BxPC-3? An IC₅₀ of 20 μM is still relatively potent. Does this mean 6a is targeting other cellular molecules? Is 6a inherently cytotoxic in and of itself? What is the rationale for this? The authors could add a non-reactive version of 6a (perhaps 6a with a methyl group instead of the Cl) and observe that cell death as a control.
- No statistical analysis is provided for Fig 5f and/or Fig S4b.
- References should utilize ISO4 journal abbreviations
- R_f values should be provided for all novel compounds.

Reviewer #3

(Remarks to the Author)

In this work, Zhang et al. described targeted covalent protein modification at glutamates with haloalkane warheads. The authors selected PDE as the target protein and synthesized several reactive ligands by attaching chloro- or bromoalkanes to reversible PDE inhibitors. In vitro assays, compounds 6a and 5e were identified as promising candidates for covalent inhibition. Mass spectrometry and crystallographic analysis revealed that these compounds unambiguously react with the E88 residue of PDE to form ester bonds. Furthermore, a series of analyses including TPP, CETSA, and gel shift assays demonstrated that 6a covalently engages PDE in cellular environments and inhibits the proliferation of cancer cells. The results represent a potential breakthrough in the development of Glu/Asp-targeted covalent inhibitors, an area where successful examples remain limited, and this study is expected to provide valuable insights to the research community. This

is overall a good paper that is well thought out and decently performed. However, the following concerns regarding stabilities of the probe and reaction products, reaction kinetics, and generality of the strategy should be addressed prior to publication.

1. From an organic chemistry standpoint, haloalkanes are relatively stable in water but reactive toward thiols. However, the authors have reported the opposite result. They have quantified the remaining probe in aqueous buffer using HPLC, but did they confirm the hydrolysis products of the probe or its reaction products with GSH? Hydrophobic compounds often aggregate in aqueous solution and are lost, which can lead to a time-dependent decrease in HPLC signal independent of degradation. To address this concern, the authors should include the raw HPLC data in the Supplementary Information (SI).
2. To clarify the irreversibility of PDE inhibition by the probe, the stability of the ester bond formed between PDE and the probe should be evaluated in aqueous conditions and in the presence of GSH.
3. The authors should report the second-order rate constant, which is a key parameter for characterizing the performance of an irreversible inhibitor, at least for 6a.
4. Figure 4: Why is there no clear 6a concentration dependency in the anisotropy assay under conditions where covalent inhibition is expected (e.g., 62.5 nM shows stronger inhibition than 125 nM)? Also, which two data points were compared to calculate the p-value? If multiple group comparisons were performed, an appropriate statistical method should be used.
5. Fig 5d,e: Using this figure—or conditions under which the two bands are more clearly separated—would it be possible to estimate the efficiency of covalent bond formation? At first glance, it appears that nearly 100% of the covalent modification occurs within 2 hours. This seems inconsistent with the reaction kinetics shown in vitro, making it an intriguing result. To further characterize the reaction properties of compound 6a, it would be better to clarify the time profile of the reaction with 6a in cells.
6. The weakest point of this study is the lack of evidence supporting the general applicability of the strategy. Can irreversible inhibition via bromoalkanes be applied to other proteins as well? For publication in a high-impact journal such as Nature Communications, I believe it is necessary to demonstrate that this strategy can be applied to at least one additional target protein.

Reviewer #4

(Remarks to the Author)

Traditional covalent inhibitors primarily target highly reactive cysteine (Cys) residues, while aspartic acid (Asp) and glutamic acid (Glu) have been less studied due to their weak nucleophilicity and the instability of the resulting ester linkages. This study focuses on covalent protein modification technology targeting glutamic acid (Glu), inspired by HaloTag technology, to develop a novel bromoalkene-warhead covalent inhibitor (DeltaTag) that specifically modifies Glu88 (E88) in the lipoprotein chaperone protein PDE δ .

The designed compounds incorporate bromoalkane warheads, and structural optimizations—such as replacing amides with sulfonamides and introducing a piperidine anchoring group—enhance binding efficiency and selectivity for PDE δ . The lead compound, DeltaTag (6a), achieves 95% covalent modification efficiency in 24 hours, with crystal structures confirming ester bond formation with E88. Functional studies demonstrate that DeltaTag overcomes Arl2/3-mediated inhibitor release, stably binding PDE δ ($\Delta T_m > 15^\circ\text{C}$), when the DeltaTag was first incubated with the PDE δ at 37°C. In cells, it inhibits the mTORC1 signaling pathway (reducing S6P phosphorylation) and suppresses the proliferation of KRas-dependent cancer cells with IC₅₀ about 4-10 μM level (e.g., pancreatic cancer PA-TU-8902).

Overall the study is interesting for the chemical biology researchers, and provided a new strategy to covalently targeting the Glu residues. However, there are also some important issues need to be addressed to improve the utility of this method.

1. The study solely focused on the E88 modification of PDE δ , lacking experimental validation of other potential targets (such as proteins with Glu/Asp in similar hydrophobic pockets, as mentioned by authors the E136 of UNC119A). Although the article proposed that this strategy might be generalizable, no actual data were provided to support the claim. The authors need to assay the Glu/Asp residues in different solvent (different hydrophobicity?) to check how microenvironmental difference affects the reactivity of Glu/Asp.
2. Although mass spectrometry did not detect modifications at other sites in PDE δ , a full proteomic analysis (not TPP method, which only analyzes the thermal effect of proteins, not the residue modification) was not performed to rule out off-target effects.
3. The study lacked validation in animal models (e.g., tumor xenografts) to assess anticancer efficacy, and the druggability and in vivo toxicity (e.g., hepatotoxicity) of the bromoalkane warhead were not evaluated.
4. Although decreased S6P phosphorylation was observed, there was no direct evidence demonstrating that DeltaTag functions by inhibiting the PDE δ -Rheb-mTORC1 axis (e.g., missing Rheb membrane localization assays).
5. The study only verified Arl2's inability to release DeltaTag via fluorescence polarization but did not analyze whether Arl2/3 binding induces PDE δ conformational changes that affect the DeltaTag binding, as the covalent binding of DeltaTag seems sensitive to the temperature.

Version 1:

Reviewer comments:

Reviewer #1

(Remarks to the Author)

This revision only addressed some of the concerns raised in the initial review. However, after careful evaluation of your responses and the additional data provided, I remain unconvinced regarding the central claims and broader significance of this study. The concerns about the advantages using such a covalent design strategy on PDE δ have not been solved. The broad applicability of the HaloTag technology is doubtful. We do not think this revision made significant improvement. The following critical issues persist:

1. Although the authors have provided preliminary data on UNC119 modification (Supplementary Fig. S6 and Table T3), these results remain highly preliminary and lack critical structural or functional validation of selective covalent modification. The fragment-sized alkyl bromide ligand demonstrated reactivity with multiple solvent-exposed acidic residues; this lack of selectivity undermines the claim of a "broadly applicable" strategy. Therefore, unless selective covalent modification is rigorously demonstrated for at least one additional protein target beyond PDE δ using a precisely designed ligand, the proposed methodology should currently be regarded as a proof-of-concept case study rather than a mature and widely generalizable technology. Furthermore, in response to comment #8, the authors emphasized the innovativeness of their covalent strategy targeting glutamate. However, given that the HaloTag technology itself already successfully utilizes aspartate (D106) for specific covalent conjugation, the core innovation and biological significance of this work—applying similar chloroalkane chemistry to PDE δ inhibitor development—should reside more in the development of a covalent inhibitor for a specific target (PDE δ) and its functional consequences (e.g., overcoming Arl2-mediated release, modulating mTOR signaling, inhibiting cancer cell proliferation), rather than overemphasizing the novelty of the covalent mechanism itself.
2. Although the authors conducted a Comet assay to rule out DNA alkylation, this does not comprehensively address the potential for off-target protein modifications within the complex proteome. Data from global, residue-level profiling techniques (e.g., activity-based protein profiling (ABPP) or chemoproteomics) to assess selectivity are still lacking. In the absence of such critical data, the claim of "selective targeting" of p.E88 in PDE δ by compound 6a remains inadequately supported, particularly given the inherent potential reactivity of haloalkane warheads.
3. The cellular efficacy of DeltaTag (6a) remains modest, with IC₅₀ values in the micromolar range for most tested cancer cell lines. More importantly, despite the editors' leniency, the lack of *in vivo* validation (e.g., in tumor xenograft models) to assess anticancer efficacy severely limits the translational potential and biological impact of the findings. The observed antiproliferative effects could still be attributable to non-specific alkylation or off-target effects rather than specific PDE δ inhibition. Furthermore, although the authors have endeavored to clarify and correct specific NMR assignments and purity concerns, the fact that errors and ambiguities existed initially in the characterization data for multiple compounds inevitably raises persistent concerns regarding the overall rigor of the analytical validation.

Reviewer #2

(Remarks to the Author)

The authors have significantly revised the manuscript and provided thorough responses to all reviewer comments. Their revisions satisfactorily address all of my previous concerns, and the overall clarity and scientific rigor of the work have improved. I believe the manuscript is now suitable for publication without further changes.

Reviewer #3

(Remarks to the Author)

The authors have addressed all my concerns, and I believe the manuscript has been improved to be published.

Reviewer #4

(Remarks to the Author)

The manuscript entitled "Covalent Modification of a Glutamic Acid Inspired by HaloTag Technology" describes the rational design and characterization of covalent PDE δ inhibitors that selectively target a glutamic acid (E88) residue. Inspired by the HaloTag system, the authors developed DeltaTag, a prototype compound capable of covalently modifying PDE δ , thereby disrupting the PDE δ –Rheb–mTORC1 signaling axis and impairing cancer cell proliferation. The study is comprehensive, integrating chemical synthesis, structural biology, proteomics, and cellular functional assays. Overall, the work represents a significant advance in expanding the scope of covalent inhibition strategies beyond cysteine/lysine residues, and it is of broad interest to the chemical biology and drug discovery communities. I believe it is suitable for publication in *Nature Communications* after major revisions.

Comments:

1. While the authors briefly demonstrate preliminary covalent modification of UNC119, more extensive validation on additional targets would strengthen the claim that the approach is broadly applicable. Alternatively, providing additional structural/biochemical evidence on another protein target with a buried Glu/Asp residue.
2. Although DNA alkylation and general cytotoxicity were tested, broader proteome-wide reactivity profiling (e.g., using

chemoproteomics with activity-based probes) could further substantiate selectivity. At minimum, discuss potential risks of non-specific reactivity in more detail, especially given the electrophilic nature of alkyl bromides.

3. The manuscript demonstrates that DeltaTag overcomes limitations of reversible PDE δ inhibitors, but the comparison remains somewhat qualitative.

4. The covalent modification is demonstrated at E88, but it remains unclear why this residue reacts preferentially over others. Consider including computational modelling or additional mutagenesis data (e.g., E88Q, E89Q) to further validate selectivity.

5. The absence of in vivo evaluation is a limitation. However, the authors could strengthen the translational relevance by reporting preliminary pharmacokinetic stability (microsomal stability, plasma stability, or logD) of DeltaTag.

Version 2:

Reviewer comments:

Reviewer #4

(Remarks to the Author)

Since the authors address all my questions, I think the current manuscript is suitable for publishing in NC.

Reviewer #1 (Remarks to the Author):

The authors designed and synthesized series of covalent PDE δ inhibitors with alkyl bromide warheads using HaloTag Technology. Among them, the compound DeltaTag (6a) shows the best performance. It can overcome Arl2-mediated release, regulate signal transduction through the mTOR signaling pathway, and inhibit cancer cell proliferation. However, this study suffers from several significant shortcomings.

Most haloalkyl drugs, such as nitrogen mustard alkylating agents, exert their antitumor effects by targeting DNA, disrupting its structure and function, and ultimately preventing cell division and proliferation. These drugs are associated with substantial toxicity. In this study, the authors failed to address the potential off-target interaction between DeltaTag and DNA or evaluate the resulting toxic side effects.

In order to constructively address the reviewers' comment, we conducted cytotoxicity screening for **6a** in a panel of human cell lines (Fig. S5c), indicating that it exhibited no general toxicity at concentrations up to 25 μ M. We did not observe potential off-target effects such as DNA alkylation by **6a** at 10 μ M in a Comet assay (Supplementary Fig. S5b) or morphological profiling at concentrations up to 30 μ M, in comparison to known DNA alkylating agents such as chlorambucil, ifosfamide and melphalan (Supplementary Fig. S5d).

Target engagement of DeltaTag in cells was supported by TPP/CETSA analysis (Fig. 4) and comparison of cell viability in a pair of HAP1 wild type and HAP1 PDE knock out cell lines (Supplementary Fig. S5a).

Furthermore, while the hypothesis that "reversible PDE δ inhibition is efficiently counterbalanced by Arl2/3-mediated inhibitor release calling for covalent inhibitor development" forms the foundation of this study, the improvement in the antitumor activity of the compound DeltaTag is extremely limited. Moreover, its observed antitumor activity may be directly attributable to its alkylating properties rather than specific PDE δ inhibition.

Our work focuses on development of a covalent inhibition strategy inspired by the HaloTag technology, and we utilise PDE δ as a model system for proof-of-concept. The primary goal was not to develop covalent PDE δ inhibitors to overcome Arl2/3-mediated release. Rather, the focus was on establishing a covalent modification method targeting at a specific glutamic acid. Hence significant enhancement of antitumor activity through PDE δ inhibition from a drug discovery perspective was not a central objective in this work.

As described above, we now have demonstrated that DeltaTag does not show unspecific toxicity.

Additionally, there are numerous errors with the compound structures and spectra data, raising concerns about the reliability of the findings. Therefore, this manuscript is not suitable for publication in NC.

We have revisited the manuscript and the compound and spectra descriptions. We addressed these issues under response to comment 10. Please see below for details.

Comments:

1. A major concern is whether PDE δ is a druggable target. The authors have developed numerous low

nanomolar PDE δ inhibitors, which generally showed weak antitumor potency. The covalent PDE δ inhibitors designed herein failed to address these limitations, which largely undermined the significance of this work.

As mentioned above, we focus on the establishment of a covalent inhibition strategy inspired by the HaloTag technology, utilising PDE δ as a model system for proof-of-concept. The question whether PDE δ is a druggable target is not in the focus of our work. This is a proof-of-principle investigation.

2. The authors should evaluate the interactions between the covalent PDE δ inhibitors with alkyl bromide warheads and DNA, and assess their toxicity in normal cells.

In order to address this comment constructively, we investigated potential off-target effects such as DNA alkylation in a Comet assay, and we did not observe undesired activity of DeltaTag at 10 μ M concentration (Supplementary Fig. S5b). Furthermore, the compounds exhibited no general cytotoxicity at concentrations up to 25 μ M in cell viability assays and morphological profiling in a panel of human cell lines (Supplementary Fig. S5c,d).

Please see page 21 of the manuscript for details.

3. Figure 4. The authors should rule out the influence of temperature. Additional experimental group with compound 6a incubated at 37°C should be included.

As requested by the reviewer we included additional experimental controls with the compound incubated at 37 °C. Please see Fig. 4b, 4c, Supplementary Fig. S3d for details.

4. Figure 4. The fluorescence polarization method is unsuitable for assessing Arl2-mediated release due to the competitive binding relationship between FA and PDE δ inhibitors. The observed decrease in anisotropy values with increasing concentrations of compound 6a directly reflects this competitive binding. Moreover, the lack of concentration dependence in the Arl2-treated groups further undermines the validity of this assay.

We agree that at reversible binding stage, the fluorescence polarisation method is not a straightforward model for monitoring Arl2 release, for the FA probe is also released by the Arl2 (Supplementary Fig. S3b). Direct comparison between conditions in the presence and the absence of Arl2 in the fluorescence polarisation assay was not feasible; however, we were able to compare reversible and covalent inhibitors under the same assay conditions. We have revised the discussion and pairwise comparisons to focus on conditions where covalency was expected for compound **6a** (Fig. 4a, 4c and S3d).

Please see the figure below for the same set of anisotropy data plotted against concentrations over the range of 0 – 500 nM, with curves fitted to a sigmoidal dose-response model. A clear dose-response relationship was not evident through direct comparison of the two concentrations, as the response did not follow a linear pattern.

Competitive fluorescence polarisation assay

5. The authors should verify whether the compound DeltaTag can overcome the Arl2-induced release in cells.

The primary goal of our work is not to develop covalent PDE δ inhibitors to overcome Arl2/3-mediated release. Instead, we introduce a covalent inhibition strategy using haloalkane warheads, with PDE δ serving as an inaugurating example. Covalent labelling was demonstrated in cells, while the impact on Arl2-mediated release was not central to our work. However, our results do provide indirect evidence in cells that DeltaTag binds covalently despite the counteracting action of Arl2 (Fig. 4). This is clearly demonstrated by sustained downstream effects along the PDE δ -Rheb-mTORC1 axis (Fig. 5).

6. Figure 5A. The interaction between small molecules and proteins may either enhance or reduce protein thermal stability. The authors exclusively considered increased stability while ignoring potential destabilization effects that could equally indicate DeltaTag interactions.

In the TPP analysis, PDE δ was the only significant hit ($p \leq 0.05$) identified that showed target stabilisation with the largest change in area under the curves (ΔAUC) as compared to treatment with DMSO ($\Delta AUC = 3.5$, p -value = 0.008, Fig. 4j). Potential proteins showing destabilisation effects upon DeltaTag treatment are highlighted as blue dots in Fig. 4j. Please refer to page 14-15 of the manuscript and Supplementary Table T2 for details.

7. In Figure S4, the antitumor activity of the covalent inhibitor 6a is not improved compared to the reversible inhibitor 6b. Notably, both 6a and 6b, as haloalkyl compounds, exhibit significantly superior antitumor activity compared to the PDE δ inhibitor Deltazinone 1. This raises the possibility that their observed activity may be directly related to their alkylating properties rather than specific PDE δ inhibition.

Comparing to its reversible counterpart **6b**, DeltaTag (**6a**) was consistently more potent than **6b**, with the exception in NCI-H358 cells for which both compounds exhibited similar activity (IC_{50} of **6a** = $6.3 \pm 3.5 \mu M$; IC_{50} of **6b** = $6.8 \pm 1.9 \mu M$, Fig. 6b). Thus, the reviewer's wording "the antitumor activity of the covalent inhibitor **6a** is not improved compared to the reversible inhibitor **6b**" does not reflect our data.

Comparing to Deltazinone **1**, DeltaTag (**6a**) exhibited a stronger growth inhibitory effect than Deltazinone **1**, except in PA-TU-8902 where both compounds showed comparable potency (IC_{50} of **6a** = $6.8 \pm 0.5 \mu M$ versus IC_{50} of Deltazinone **1** = $4.6 \pm 1.4 \mu M$, Fig. 6b). In NCI-H358 cells Deltazinone **1** induces significantly stronger growth inhibition (IC_{50} = $0.14 \pm 0.11 \mu M$, Fig. 6b). These findings are not adequately reflected by the reviewer's statement that "both **6a** and **6b**, as haloalkyl compounds, exhibit significantly

superior antitumor activity compared to the PDE δ inhibitor Deltazinone **1**". The enhanced cellular potency of **6a** cannot be attributed solely to its alkylating properties.

Besides TPP/CETSA (Fig. 4) and the observed impact on the PDE δ -Rheb-mTORC1 axis (Fig. 5), in-cell target engagement was further supported by the observation that growth impairment of HAP1 PDE δ knockout cells by DeltaTag was less pronounced compared to HAP1 wild type cells at concentrations of 0 - 50 μ M (paired *t*-test, two-tailed *p*-value = 0.049, Supplementary Fig. S5a).

Potential off-target effects such as DNA alkylation were not observed with DeltaTag treatment at 10 μ M in a Comet assay (Supplementary Fig. S5b). Furthermore, the compounds (**6a** and **6b**) exhibited no general cytotoxicity at concentrations up to 25 μ M, as demonstrated by cell viability and morphological profiling in a panel of human cell lines (Supplementary Fig. S5c,d).

Please refer to results, page 20-21 of the manuscript and Fig. 6 for details.

8. The covalent inhibitors were not evaluated in animal models. More specifically, pharmacokinetic profiles, *in vivo* antitumor potency and *in vivo* toxicity remain to be assayed, which is important to confirm the potential advantages of covalent inhibitors.

The purpose of our work is to establish a covalent inhibition strategy with PDE δ as a model system and the best covalent inhibitor DeltaTag characterised in greater detail under biologically relevant conditions. The pharmacokinetic profiles, *in vivo* potency and toxicity of the covalent inhibitors, however, are not within the scope of this study and, therefore, were not evaluated with a view to drug discovery. Please see also our explanations above.

In the decision letter the Editors informed us that they "... do not require *in vivo* validation of antitumor activity of covalent inhibitors (as asked by Reviewers 1 and 4)."

9. In Figure 6, the authors have made theoretical predictions but have not provided direct experimental evidence to support the broader applicability of this approach.

Please see our discussion at page 23-24 of the manuscript. To explore the broader applicability of alkyl bromides as covalent warheads, we adopted a fragment-based approach to design an alkyl bromide ligand targeting UNC119 protein and a specific glutamic acid in its lipid-binding site. Experimental evidence supporting covalent modifications at acid residues by the ligand is provided in Supplementary Fig. S6 and Supplementary Table T3.

10. There are numerous errors in the compound characterization data (Supplementary Materials). The details are listed in the review attachment.

From the review attachment,

- a) Several compounds (3a, 3c, 3d, 4a, 4b, 4c, 4d, 5a, 5b, 5c, 5d, 5e, 5f, 5g, 5h, 5i, 5j, 5k, 5l, 6a, 6b) exhibit multiple unexplained signals at δ 8.0–9.0 in their ¹H NMR spectra, suggesting the presence of consistent impurities and indicating inadequate compound purity.

As indicated in the compound characterisation, these compounds (**3a**, **3c**, **3d**, **4a**, **4b**, **4c**, **4d**, **5a**, **5b**, **5c**, **5d**, **5e**, **5f**, **5g**, **5h**, **5i**, **5j**, **5k**, **5l**, **6a**, **6b**) were purified by prep-HPLC and characterised as TFA salts. The two broad singlets at δ 8.0–9.0 in the proton NMR correspond to the $-\text{NH}_2^+$ protons and are not an indication of impurities. We thank the reviewer for addressing the discrepancy in the structures drawn. We have corrected the structures for all compounds. An example of ^1H NMR peak listing is shown below for compound **3a**.

***N*-(3-chloropropyl)-4-(3,4-dimethyl-7-oxo-2-(*p*-tolyl)-2,7-dihydro-6*H*-pyrazolo[3,4-*d*]pyridazin-6-yl)-*N*-(piperidin-4-ylmethyl)butanamide (3a)**

^1H NMR (600 MHz, $\text{DMSO-}d_6$) δ 9.03 (br. s, 1H), 8.71 (br. s, 1H), 7.44 (d, $J = 8.1$ Hz, 2H), 7.40 (d, $J = 8.1$ Hz, 2H), 4.02 (t, $J = 7.1$ Hz, 2H), 3.87 (4H hiding in water peak), 3.23 (d, $J = 13.3$ Hz, 2H), 2.94 (t, $J = 6.3$ Hz, 2H), 2.81 (q, $J = 12.6$ Hz, 2H), 2.56 (s, 3H), 2.49 (s, 3H), 2.39 (s, 3H), 2.13 (t, $J = 7.6$ Hz, 2H), 1.97 – 1.87 (m, 2H), 1.79 – 1.69 (m, 2H), 1.71 – 1.61 (m, 1H), 1.33 – 1.23 (m, 2H), 1.23 – 0.98 (m, 2H).

b) Page 39. The ^1H NMR spectrum of compound **2a** does not match its proposed structure (missing a CH_2 group)

We reanalysed compound **2a** for ^1H NMR and provided the following ^1H NMR assignment.

***N*-(3-chloropropyl)-4-(3,4-dimethyl-7-oxo-2-(*p*-tolyl)-2,7-dihydro-6*H*-pyrazolo[3,4-*d*]pyridazin-6-yl)butanamide (2a)**

^1H NMR (700 MHz, $\text{DMSO-}d_6$) δ 7.48 (br. s, 1H, amide NH), 7.48 (d, $J = 8.3$ Hz, 2H, aromatic H), 7.43 (d, $J = 8.4$ Hz, 2H, aromatic H), 4.08 (t, $J = 6.9$ Hz, 2H), 4.02 (t, $J = 6.4$ Hz, 2H), 3.65 (t, $J = 6.6$ Hz, 2H, CH_2Cl), 2.59 (s, 3H, CH_3), 2.54 (s, 3H, CH_3), 2.43 (s, 3H, CH_3), 2.35 (t, $J = 7.4$ Hz, 2H), 1.99 – 1.93 (m, 2H), 1.87 – 1.77 (m, 2H).

c) Page 94. The ^{13}C spectrum of compound **6b** contains unassigned peaks at δ 176, 77, and 67, indicating potential impurities or low purity.

Purity of compound **6b** was checked by HPLC. Please see below for HPLC traces.

d) ^1H NMR spectra for compounds 2a, 2b, 2c, and 2d lack signals corresponding to -NH groups. For **2a-2d**: A broadened singlet at 7.48 ppm (br. s, 1H) corresponding to the amide NH group is found to merged with the aromatic doublet hydrogens. In the presence of water in DMSO- d_6 , signals corresponding to the exchangeable NH protons are broadened. We have corrected the integrations and included this peak for assignment.

e) The ^1H NMR assignment for compound S5 is incorrect, with the carboxyl (-COOH) peak missing from the reported data. We thank the reviewer for pointing out the missing carboxyl peak. We have added the broad singlet at 12.08 ppm to the proton NMR assignment.

4-(3,4-Dimethyl-7-oxo-2-(*p*-tolyl)-2,7-dihydro-6H-pyrazolo[3,4-*d*]pyridazin-6-yl)butanoic acid (S5)

¹H NMR (700 MHz, DMSO-*d*₆) δ 12.08 (br. s, 1H), 7.48 (d, *J* = 8.2 Hz, 2H), 7.43 (d, *J* = 8.2 Hz, 2H), 4.07 (t, *J* = 7.0 Hz, 2H), 2.59 (s, 3H), 2.52 (s, 3H), 2.43 (s, 3H), 2.26 (t, *J* = 7.4 Hz, 2H), 1.93 (p, *J* = 7.2 Hz, 2H).

f) The ¹H NMR assignment for compound S4 contains errors: 2.52 (s, 11H).
We provided the following ¹H NMR assignment for compound S4.

Methyl 4-(3,4-dimethyl-7-oxo-2-(*p*-tolyl)-2,7-dihydro-6*H*-pyrazolo[3,4-*d*]pyridazin-6-yl) butanoate (S4)

¹H NMR (700 MHz, CDCl₃) δ 7.36 (d, *J* = 8.2 Hz, 2H, aromatic Hs), 7.33 (d, *J* = 8.2 Hz, 2H, aromatic Hs), 4.26 (t, *J* = 6.9 Hz, 2H), 3.66 (s, 3H, OCH₃), 2.62 (s, 3H, CH₃), 2.58 (s, 3H, CH₃), 2.45 (s, 3H, CH₃), 2.43 (t, *J* = 8.0 Hz, 2H), 2.17 (p, *J* = 7.2 Hz, 2H, CH₂CH₂CH₂).

Reviewer #2 (Remarks to the Author):

Publication with minor revisions

Zhang and co-workers have developed a covalent inhibitor for PDEδ, DeltaTag, based off inspiration from chloroalkane system. They utilize an alkyl bromide handle to selectively react with a glutamic acid deep in the active site of PDEδ, and displayed multiple downstream effects of this inhibition in live cells. Overall, the work is thorough and robust and a good contribution to Nature Communications, though the authors need to address the following concerns.

We thank the reviewer for the positive evaluation.

- The title of the paper, caption of abstract figure, and major portions of the introduction discuss bioconjugation techniques to Asp/Glu. This paper modifies only one Asp/Glu in PDEδ, and is not applicable to global Asp/Glu profiling, though the title and intro suggest this this a broadly applicable approach to modify any Asp/Glu residue. Please adjust language to clarify you are only targeting one specific Glu.

We changed the title, the caption of the figure and modified the Introduction to clarify that we are targeting a specific glutamic acid, p.E88 of PDEδ.

- All individual data points should be shown in all graphs including Fig 4, Fig 5g, Fig S3, and other applicable locations.

Figures were modified to show all individual data points in graphs where applicable. Applicable locations are Fig.2b, 2e, 5b, 5f, S1b, S1c, S1d, S2a, S5b.

- It is unclear to me why 6b is reversible. While Br is a better leaving group (for 6a), I am surprised 0% of 6b (or 5h), especially since this paper is framed around the chloroalkane HaloTag which explicitly uses Cl as its leaving group.

The reactivity of alkyl chlorides for covalency in the HaloTag system is achieved through generations of protein engineering, during which key amino acid residues were selected for site specific mutagenesis to optimise the binding site and disposition of p.D106 for labelling kinetics and duration. We reasoned that without precise positioning of **6b** or **5h** in the binding pocket of PDE δ towards p.E88, no covalency was observed.

Furthermore, the IC₅₀ of 6b is ~1/2 of 6a, making this an even more interesting discovery. A discussion around this discrepancy should be added.

The reversible PDE δ inhibitor **6b** showed some growth inhibitory effect in KRAS-mutant cell lines with IC₅₀ values generally >10 μ M, therefore limiting its cellular applications. The consistent improvement in cellular potency from the reversible inhibitor **6b** to the covalent PDE δ inhibitor **6a** supported the design strategy for covalent inhibitor development. We have added this discussion of the comparison – please see page 21 of the manuscript.

- Titles of Supplementary Figures should come before their respective graphs and writing, not in the middle between the figure and the writing.

We changed the format of the Supplementary Information, with titles of Supplementary Figures, followed by Figures and figure captions.

- At the bottom of Page 8 the authors state “...which was reflected in the enhancement of thermostability from $\Delta T_m = 13.5 \pm 0.2$ °C to $\Delta T_m = 15.8 \pm 0.8$ °C...” without providing statistical context. Is this change significant? No p-values are provided in Fig 2e supporting this claim.

We conducted unpaired *t*-test comparing melting temperature shifts between PDE δ •**5e** and PDE δ •**5h** (two-tailed *p*-value = 0.0031), between PDE δ •**6a** and PDE δ •**5b** (two-tailed *p*-value = 0.0002), and between incubation with **6a** at 2 hours and 24 hours (two-tailed *p*-value = 0.0001). Changes mentioned in the text are therefore statistically significant. Statistics were added to the main text, Fig. 2e and 2f.

- Fig 2f: Why is the energy change for the T_m of PDE δ -DMSO controls almost negligible? The SI states all experiments were conducted with equal concentrations, yet the controls give ~10% of the signal from the PDE δ -6a samples?

The y-axis of Fig. 2f shows the first derivative of the fluorescence ratio at wavelength 330 nm/350 nm, as a measure of the rate of protein unfolding. Melting temperatures were recorded at the inflection points. The shape of the melting curve is an indication how sharp a protein melts, not as an indication of energy change. The curves suggest that PDE δ bound to compounds (‘closed’ form) has a relatively sharp melting point as compared to its ‘open’ form (PDE δ -DMSO controls).

- All protein intact MS figures should have the exact mass displayed about the peak. Notably in Fig 2c and Fig S3a.

Fig. 2c and Fig.S3a were modified to show the exact mass of PDEδ peaks.

- Fig S3a: The intensity units are drastically different between samples, ranging from upwards of 5000 a.u. to as low as 30 a.u. and seemingly decrease with added temperature. This is confusing as the SI states these experiments were done at the exact same concentrations. Please repeat with a similar injection volume for each data point.

These experiments were done at the same concentrations for incubation. Intensity units were measured by MALDI with accumulation of counts on linear positive ion mode. We repeated the measurements with similar accumulative intensity to 400 - 600 a.u. for each condition and included the spectra in Fig. S3a.

- Fig 5g: The use of * is for general ranges of p-values, as the authors utilized in Fig 4, not for exact p-values. Additionally, without brackets, it is difficult to determine what comparison the p-value is referring to.

We added brackets in Fig. 5f (new numbering of the figure for Western blot analysis of S6P phosphorylation) to indicate the pair-wise comparisons. Exact p-values were provided in figure legend.

Fig. 5f: “P-values were determined by applying unpaired *t*-test comparing the respective condition to EGF stimulated DMSO control (-/+). Two-tailed *p*-values, * < 0.05; ** < 0.01 and *** < 0.001, *p*-value > 0.05 are labelled as not significant (ns). 3 h: -/+ vs **1**, *p* = 0.78, vs **6a**, *p* = 0.035; 5 h: -/+ vs **1**, *p* = 0.138, vs **6a**, *p* = 0.0009; 8 h: -/+ vs **1**, *p* = 0.102, vs **6a**, *p* < 0.0001.”

- Bottom of Page 16: The authors state “Overall, 6a,b exhibited a stronger growth inhibitory effect than Deltazinone 1 in these cell lines with the exception of NCI-H358”. This is false, as Fig S3 states that in cell line PA-TU-8902, 1 has lower IC₅₀ than either 6a or 6b. The authors should discuss why 1 is significantly better for PA-TU-8902 and NCI-H358, but had almost no effect on the other oncogenic KRas dependency lines (MIA-PaCa-2 and SW-480)

We changed the discussion and the statement that “DeltaTag (**6a**) exhibited a stronger growth inhibitory effect than Deltazinone **1**, except in PA-TU-8902 where both compounds showed comparable potency (IC₅₀ of **6a** = 6.8 ± 0.5 μM versus IC₅₀ of Deltazinone **1** = 4.6 ± 1.4 μM, Fig. 6b). In NCI-H358 cells where Deltazinone **1** induces significantly stronger growth inhibition (IC₅₀ = 0.14 ± 0.11 μM, Fig. 6b).”

We observed exceptions to the general trend of cellular potency in NCI-H358 cells for the tested compounds. We speculate that this discrepancy might be related to tissue specificity, KRAS mutational status, and/or the degree of dependency on the target pathway, though the exact cause remains unclear.

- Why is 6a effective at growth inhibition for non-oncogenic KRas dependent cell line BxPC-3? An IC₅₀ of 20 uM is still relatively potent. Does this mean 6a is targeting other cellular molecules? Is 6a inherently

cytotoxic in and of itself? What is the rationale for this? The authors could add a non-reactive version of 6a (perhaps 6a with a methyl group instead of the Cl) and observe that cell death as a control.

In-cell target engagement to PDE δ was supported by the observation that growth impairment of HAP1 PDE δ knockout cells by 6a was less pronounced compared to HAP1 wild type cells at concentrations of 0 - 50 μ M (paired *t*-test, two-tailed *p*-value = 0.049, Supplementary Fig. S5a).

We conducted cytotoxicity screening for 6a in a panel of human cell lines (Fig. S5c), indicating that it exhibited no general toxicity at concentrations up to 25 μ M. We did not observe potential off-target effects such as DNA alkylation by 6a at 10 μ M in a Comet assay (Supplementary Fig. S5b) or morphological profiling at concentrations up to 30 μ M (Supplementary Fig. S5d).

An IC₅₀ of 20.0 μ M for 6a in BxPC-3 cells was at the borderline of unspecific toxicity. The difference of cellular potency exhibited by 6a in a KRAS-dependent cell line as compared to a KRAS-independent cell line suggested a correlation between PDE δ inhibition to its downstream client protein KRAS, by analogy to Deltazinone 1.

- No statistical analysis is provided for Fig 5f and/or Fig S4b.

We added statistical analysis where applicable. Applicable locations are Fig. 2e, 2f, 5b, 5f, 6c, 6d, S1d, S5a, S5b.

- References should utilize ISO4 journal abbreviations.

We changed references to utilise abbreviations for journals.

- Rf values should be provided for all novel compounds.

Rf values were added – please see Supplementary Information, compound characterisation.

Reviewer #3 (Remarks to the Author):

In this work, Zhang et al. described targeted covalent protein modification at glutamates with haloalkane warheads. The authors selected PDE δ as the target protein and synthesized several reactive ligands by attaching chloro- or bromoalkanes to reversible PDE δ inhibitors. In in vitro assays, compounds 6a and 5e were identified as promising candidates for covalent inhibition. Mass spectrometry and crystallographic analysis revealed that these compounds unambiguously react with the E88 residue of PDE δ to form ester bonds. Furthermore, a series of analyses including TPP, CETSA, and gel shift assays demonstrated that 6a covalently engages PDE in cellular environments and inhibits the proliferation of cancer cells. The results represent a potential breakthrough in the development of Glu/Asp-targeted covalent inhibitors, an area where successful examples remain limited, and this study is expected to provide valuable insights to the research community. This is overall a good paper that is well thought out and decently performed. However,

the following concerns regarding stabilities of the probe and reaction products, reaction kinetics, and generality of the strategy should be addressed prior to publication.

We thank the reviewer for the positive evaluation and the suggestions for improvement.

1. From an organic chemistry standpoint, haloalkanes are relatively stable in water but reactive toward thiols. However, the authors have reported the opposite result. They have quantified the remaining probe in aqueous buffer using HPLC, but did they confirm the hydrolysis products of the probe or its reaction products with GSH? Hydrophobic compounds often aggregate in aqueous solution and are lost, which can lead to a time-dependent decrease in HPLC signal independent of degradation. To address this concern, the authors should include the raw HPLC data in the Supplementary Information (SI).

We added a set of HPLC-MS spectra to Supplementary Fig. S2b, showing stability of **6a** incubated in phosphate buffer with GSH after 72 hours at 37 °C. The MS spectra confirmed the formation of GSH adduct, phosphate adduct, hydrolysed product and chloro-substituted product of **6a**.

2. To clarify the irreversibility of PDE inhibition by the probe, the stability of the ester bond formed between PDE and the probe should be evaluated in aqueous conditions and in the presence of GSH.

We incubated the formed covalent PDE δ adducts in the following aqueous buffers for 3 days at 37 °C and did not observe any reversal to unmodified protein.

Buffers:

1. 20 mM HEPES, pH = 7.5, 150 mM NaCl;
2. 0.2 M Na Acetate buffer, pH = 5.6;
3. 0.2 M Citric buffer, pH = 6.2;
4. 0.2 M Phosphate buffer, pH = 7.4, 1 mM EDTA;
5. 0.2 M Phosphate buffer, pH = 7.4, 1 mM EDTA, 10 mM GSH.

This finding is in agreement with the previously reported stability of an ester bond formed with PDE δ in the presence of 50-fold excess of hydroxylamine and release factor Arl2. We have added this result indicating irreversibility of the ester bond to page 8 of the manuscript.

3. The authors should report the second-order rate constant, which is a key parameter for characterizing the performance of an irreversible inhibitor, at least for **6a**.

We added the modelling and calculation of the second-order rate constant for **6a** to Supplementary Fig. S1c. With **6a** only showing moderate labelling kinetics to PDE δ , the published model for relative potent binders and hence a global fit over a range of concentrations to obtain k_{inact}/K_I was not applicable.^{1,2} We approximated k_{obs} values with pseudo-first kinetics and obtained separate values for k_{inact} and K_I by plotting k_{obs} against concentrations and fitted to the Michaelis-Menten model ($k_{obs} = \frac{k_{inact} * x}{K_I + x}$, x = conc.).

We obtained a $k_{inact} = 52.55 \text{ h}^{-1}$ and a K_I value of 464.9 μM (goodness of fit, $R^2 = 0.981$). The second-order rate constant k_{inact}/K_I was calculated to be $0.113 \mu\text{M}^{-1}\text{h}^{-1} = 31 \text{ M}^{-1}\text{s}^{-1}$ (Fig. S1c).

References:

- 1 Sirocchi, L. S. *et al.* Discovery of Carbodiimide Warheads to Selectively and Covalently Target Aspartic Acid in KRASG12D. *J. Am. Chem. Soc.* **147**, 15787-15795, doi:10.1021/jacs.5c03562 (2025).
- 2 Li, K. S. *et al.* High-Throughput Kinetic Characterization of Irreversible Covalent Inhibitors of KRASG12C by Intact Protein MS and Targeted MRM. *Anal. Chem.* **94**, 1230-1239, doi:10.1021/acs.analchem.1c04463 (2022).

4. Figure4: Why is there no clear 6a concentration dependency in the anisotropy assay under conditions where covalent inhibition is expected (e.g., 62.5 nM shows stronger inhibition than 125 nM)?

Please see the figure below for the same set of anisotropy data plotted against concentrations over the range of 0 – 500 nM, with curves fitted to a sigmoidal dose-response model. A clear dose-response relationship was not evident through direct comparison of the two concentrations, as the response did not follow a linear pattern.

Competitive fluorescence polarisation assay

Also, which two data points were compared to calculate the p-value? If multiple group comparisons were performed, an appropriate statistical method should be used.

We included additional experimental controls to rule out the influence of temperature and chose to present pairwise comparisons in separate sets of figures (Fig. 4b, 4c, S3d) and to compare IC₅₀ values between conditions.

5. Fig5d,e: Using this figure—or conditions under which the two bands are more clearly separated—would it be possible to estimate the efficiency of covalent bond formation? At first glance, it appears that nearly 100% of the covalent modification occurs within 2 hours. This seems inconsistent with the reaction kinetics shown in vitro, making it an intriguing result. To further characterize the reaction properties of compound 6a, it would be better to clarify the time profile of the reaction with 6a in cells.

We added the time profile of the PDEδ labelling reaction by 6a to Fig. 4g. Consistently from 3 biological replicates, we estimated 100% covalent modification of PDEδ in 50 μg of Jurkat cell lysates by 10 μM treatment of 6a.

For the reaction kinetics shown *in vitro* (Fig. 2d), we set up the assay with compound to protein ratio as 3 to 1. In cells, we expected cellular PDE δ protein concentration in the nanomolar range. Hence treatment of cells by 10 μ M of **6a** fit to a pseudo-first reaction kinetic of covalent modification (see Figure below).

Simple linear regression with constrain to (0,0), slope = $0.8927 \pm 0.0758 \text{ min}^{-1}$

From the in-lysate labelling kinetics, we estimated a $k_{\text{obs}} = 0.8927 \text{ min}^{-1} = 53.56 \text{ h}^{-1}$

We added additional *in vitro* labelling experiments to estimate the apparent second-order rate constant of **6a** (Supplementary Fig. S1c). We obtained a $k_{\text{inact}} = 52.55 \text{ h}^{-1}$ from the *in vitro* model, which is comparable to the k_{obs} value estimated from the in-lysate labelling kinetics.

6. The weakest point of this study is the lack of evidence supporting the general applicability of the strategy. Can irreversible inhibition via bromoalkanes be applied to other proteins as well? For publication in a high-impact journal such as Nature Communications, I believe it is necessary to demonstrate that this strategy can be applied to at least one additional target protein.

We constructively addressed this comment. Please see our discussion at page 23-24 of the manuscript. To explore the broader applicability of alkyl bromides as covalent warheads, we adopted a fragment-based approach to design a ligand targeting UNC119 protein and a specific glutamic acid in its lipid-binding site. Experimental evidence supporting covalent modifications by the ligand is provided in Supplementary Fig. S6 and Supplementary Table T3.

Reviewer #4 (Remarks to the Author):

Traditional covalent inhibitors primarily target highly reactive cysteine (Cys) residues, while aspartic acid (Asp) and glutamic acid (Glu) have been less studied due to their weak nucleophilicity and the instability of the resulting ester linkages. This study focuses on covalent protein modification technology targeting glutamic acid (Glu), inspired by HaloTag technology, to develop a novel bromoalkene-warhead covalent inhibitor (DeltaTag) that specifically modifies Glu88 (E88) in the lipoprotein chaperone protein PDE δ .

The designed compounds incorporate bromoalkane warheads, and structural optimizations—such as replacing amides with sulfonamides and introducing a piperidine anchoring group—enhance binding efficiency and selectivity for PDE δ . The lead compound, DeltaTag (**6a**), achieves 95% covalent modification efficiency in 24 hours, with crystal structures confirming ester bond formation with E88. Functional studies demonstrate that DeltaTag overcomes Arl2/3-mediated inhibitor release, stably binding PDE δ ($\Delta T_m > 15^\circ\text{C}$), when the Deltatag was first incubate with the PDE δ at 37C. In cells, it inhibits the

mTORC1 signaling pathway (reducing S6P phosphorylation) and suppresses the proliferation of KRas-dependent cancer cells with IC50 about 4-10 μ M level (e.g., pancreatic cancer PA-TU-8902).

Overall, the study is interesting for the chemical biology researchers, and provided a new strategy to covalently targeting the Glu residues. However, there are also some important issues need to be addressed to improve the utility of this method.

We thank the reviewer for the positive comments.

1. The study solely focused on the E88 modification of PDE δ , lacking experimental validation of other potential targets (such as proteins with Glu/Asp in similar hydrophobic pockets, as mentioned by authors the E136 of UNC119A). Although the article proposed that this strategy might be generalizable, no actual data were provided to support the claim. The authors need to assay the Glu/Asp residues in different solvent (different hydrophobicity?) to check how microenvironmental difference affects the reactivity of Glu/Asp.

(Remark before discussion: the numbering of the glutamic acid in the lipid-binding pocket of UNC119A is E163 according to the protein sequence of PDB code 5L7K.)

We constructively addressed the reviewer's comment. Please see our discussion at page 23-24 of the manuscript. To explore the broader applicability of alkyl bromides as covalent warheads, we adopted a fragment-based approach to design a ligand targeting UNC119 protein and a specific glutamic acid in its lipid-binding site. Experimental evidence supporting covalent modifications by the ligand is provided in Supplementary Fig. S6 and Supplementary Table T3.

For PDE δ , we checked covalent modifications by DeltaTag (**6a**) in solvents with different pH values and observed that the labelling is pH-independent with no significant changes in labelling efficiency (Supplementary Fig. S1d). For UNC119, without structural optimisation for binding affinity and selectivity, the fragment-sized alkyl bromide ligand reacted not only with the target glutamic acid in the binding pocket but also with solvent-exposed glutamic and aspartic acids (Supplementary Fig. S6, Supplementary Table T3). This suggests that the reactivity of Glu/Asp residues is not significantly influenced by their microenvironment.

2. Although mass spectrometry did not detect modifications at other sites in PDE δ , a full proteomic analysis (not TPP method, which only analyzes the thermal effect of proteins, not the residue modification) was not performed to rule out off-target effects.

The design strategy presented in this study specifically targeted a glutamic acid residue p.E88 in PDE δ . Therefore, we did not perform full proteomic-scale analysis of residue modifications, which would be more relevant for methods aimed at global Asp/Glu profiling. The selectivity of the inhibitors for p.E88 in PDE δ was achieved through the binding affinity of the reversible scaffold. Conversely, we demonstrated that without the predisposed affinity and selectivity of a structured scaffold, as seen with the fragment-sized bromoalkyl ligand targeting p.E174 in UNC119B, covalent modifications target carboxylic acids more broadly, affecting several surface-exposed acidic residues.

In addition to target engagement by TPP analysis, in-cell target engagement was further supported by the observation that growth impairment of HAP1 PDE δ knockout cells by DeltaTag (**6a**) was less pronounced compared to HAP1 wild type cells at concentrations of 0 - 50 μ M (paired *t*-test, two-tailed *p*-value = 0.049, Supplementary Fig. S5a). – See results, page 21 of the manuscript.

Potential off-target effects such as DNA alkylation was ruled out by a Comet assay (Supplementary Fig. S5b) and morphological profiling (Supplementary Fig. S5d). General cytotoxicity for the compounds **6a** and **6b** were not observed at concentrations up to 25 μ M in a panel of human cell lines (Supplementary Fig. S5c).

3. The study lacked validation in animal models (e.g., tumor xenografts) to assess anticancer efficacy, and the druggability and in vivo toxicity (e.g., hepatotoxicity) of the bromoalkane warhead were not evaluated.

The purpose of our work was the establishment of a covalent inhibition strategy with PDE δ as a model system, and the best covalent inhibitor DeltaTag was characterised in greater detail. The druggability of PDE δ is not central to the method's establishment, and thus, the *in vivo* toxicity of the bromoalkane warhead was not evaluated with a view to drug discovery.

In the decision letter the Editors informed us that they “.... do not require in vivo validation of antitumor activity of covalent inhibitors (as asked by Reviewers 1 and 4).”

4. Although decreased S6P phosphorylation was observed, there was no direct evidence demonstrating that DeltaTag functions by inhibiting the PDE δ -Rheb-mTORC1 axis (e.g., missing Rheb membrane localization assays).

In order to address this comment constructively, we conducted additional experiments to demonstrate the cellular effects of DeltaTag on the PDE δ -Rheb-mTORC1 axis. Please refer to pages 17-18 and Figure 5 for details. Specifically, we added Rheb membrane localization assays by monitoring changes in fluorescence lifetime in a cellular system co-transfected with mCitrine-Rheb and mCherry-PDE δ (Fig. 5a,b), as well as by immunofluorescence staining of endogenous Rheb in PA-TU-8902 cells (Fig. 5c).

5. The study only verified Arl2's inability to release DeltaTag via fluorescence polarization but did not analyze whether Arl2/3 binding induces PDE δ conformational changes that affect the DeltaTag binding, as the covalent binding of Deltatag seems sensitive to the temperature.

Arl2 modulates a conformational equilibrium for PDE δ . Out of this equilibrium the covalent reaction selects the ‘closed’ conformation. When DeltaTag (**6a**) was presented in excess, no significant changes in the rate of covalent modifications were observed in presence or in absence of Arl2•GppNHp (Fig. 4a). This finding suggests that binding of DeltaTag (**6a**) to PDE δ is not significantly disrupted by Arl2-induced release. Formation of the ester adduct at 37 °C shifts the reaction equilibrium without notable reduction in reaction rate, especially when the electrophile is present in excess.

From the fluorescence polarisation assays, we observed significant difference in anisotropy readouts. We measured IC50 values between two treatment conditions, when Arl2 was added to a reaction of **6a** and

PDE δ , and, alternatively, when Arl2 was added after overnight incubation of **6a** with PDE δ (Fig. 4c). This suggests that the conformational change of PDE δ induced by Arl2 allosteric binding negatively impacts DeltaTag binding, especially at low concentrations of DeltaTag (**6a**).

In cells, direct monitoring of Arl2 release is not feasible, as Arl2 knock out is lethal. However, indirect evidence from TPP analysis revealed destabilisation of Arl2 ($\Delta AUC = -0.97$, p -value = 0.58, $\Delta T_m = -0.92$ °C, Fig. 4j) upon DeltaTag treatment. This finding suggests that DeltaTag binding was not prevented through Arl2-mediated release, and that the formation of the covalent adduct locked PDE δ in its 'closed' form, which adversely impacts Arl2's allosteric binding.

Please see results for details - page 13 of the manuscript and Fig. 4.

REVIEWER COMMENTS

Reviewer #1 (Remarks to the Author):

This revision only addressed some of the concerns raised in the initial review. However, after careful evaluation of your responses and the additional data provided, I remain unconvinced regarding the central claims and broader significance of this study. The concerns about the advantages using such a covalent design strategy on PDE δ have not been solved. The broad applicability of the HaloTag technology is doubtful. We do not think this revision made significant improvement. The following critical issues persist:

1. Although the authors have provided preliminary data on UNC119 modification (Supplementary Fig. S6 and Table T3), these results remain highly preliminary and lack critical structural or functional validation of selective covalent modification. The fragment-sized alkyl bromide ligand demonstrated reactivity with multiple solvent-exposed acidic residues; this lack of selectivity undermines the claim of a "broadly applicable" strategy. Therefore, unless selective covalent modification is rigorously demonstrated for at least one additional protein target beyond PDE δ using a precisely designed ligand, the proposed methodology should currently be regarded as a proof-of-concept case study rather than a mature and widely generalizable technology.

We agree the Reviewer that "the proposed methodology should currently be regarded as a proof-of-concept case study rather than a mature and widely generalizable technology". We have adjusted the language in the abstract, introduction and discussion respectively.

Furthermore, in response to comment #8, the authors emphasized the innovativeness of their covalent strategy targeting glutamate. However, given that the HaloTag technology itself already successfully utilizes aspartate (D106) for specific covalent conjugation, the core innovation and biological significance of this work—applying similar chloroalkane chemistry to PDE δ inhibitor development—should reside more in the development of a covalent inhibitor for a specific target (PDE δ) and its functional consequences (e.g., overcoming Arl2-mediated release, modulating mTOR signaling, inhibiting cancer cell proliferation), rather than overemphasizing the novelty of the covalent mechanism itself.

We agree with the Reviewer that we do not intend to claim novelty of the methods, but rather it is a proof-of-concept study in the model system of PDE δ drawing parallel to the design strategy utilised in the HaloTag system. We have adjusted texts to clarity.

2. Although the authors conducted a Comet assay to rule out DNA alkylation, this does not comprehensively address the potential for off-target protein modifications within the complex proteome. Data from global, residue-level profiling techniques (e.g., activity-based protein profiling (ABPP) or chemoproteomics) to assess selectivity are still lacking. In the absence of such critical data, the claim of "selective targeting" of p.E88 in PDE δ by compound 6a remains

inadequately supported, particularly given the inherent potential reactivity of haloalkane warheads.

We agree with the Reviewer that the current data only suggest the potential of selective targeting of p.E88 in PDE δ by compound **6a** (DeltaTag), with its structure adopted the nanomolar binding affinity and target specificity of Deltazinone 1 towards PDE δ . We assessed the inherent potential reactivity of the bromoalkane warhead in compound **6a** by *in vitro* stability assay (Supplementary Fig. S2) and found that it reacted only slowly with the most reactive sulphur nucleophile glutathione ($t_{1/2}$ for GSH adduct formation = 1035 h). *In vitro* experiment with PDE δ demonstrated that the labeling was selectively achieved at its binding site p.E88, as confirmed by mass spectrometric evaluation of the digested covalent adduct with parallel search for modifications at Glu, Asp, Cys, Thr, Ser, Tyr, Lys and His residues. However, we do not claim proteome-wide selectivity for covalent targeting of p.E88 in PDE δ and we have revised the texts accordingly. Please see our additional discussion in page 23.

3. The cellular efficacy of DeltaTag (**6a**) remains modest, with IC₅₀ values in the micromolar range for most tested cancer cell lines. More importantly, despite the editors' leniency, the lack of *in vivo* validation (e.g., in tumor xenograft models) to assess anticancer efficacy severely limits the translational potential and biological impact of the findings. The observed antiproliferative effects could still be attributable to non-specific alkylation or off-target effects rather than specific PDE δ inhibition.

We agree with the Reviewer that cellular efficacy of DeltaTag (**6a**) remains modest with IC₅₀ values in the micromolar range. We and others (the Abankwa group, the Ismail group and the Sheng group) have reported and consistently observed PDE δ inhibitors with nanomolar affinity *in vitro* but only micromolar IC₅₀ *in cellulo*. Even PDE δ degraders (PMID 31829492, PMID 38229750, PMID 38229750, PMID 39146536, PMID 35127385) that can act stoichiometrically to PDE δ and examples with demonstrated *in vivo* efficacy against KRAS mutant primary cell line derived xenografts (PMID 38229750, PMID 35127385) showed with micromolar IC₅₀ *in cellulo*. We do not want to speculate any reasoning.

The link between the observed antiproliferative effects and on-target inhibition of PDE δ was supported by the cell viability assay in the PDE δ knock out cell line (Supplementary Fig. S5a). We observed that growth inhibition of HAP1 PDE δ knockout cells by DeltaTag was much less pronounced compared to HAP1 wild type cells at concentrations of 0 - 50 μ M (paired *t*-test, two-tailed *p*-value = 0.049, Supplementary Fig. S5a) and discussed this result in the manuscript, page 21, L489-492.

We do not intend to claim translational potential of the findings presented in this study for anti-cancer activity. We, however, performed additional experiments to assess preliminary physiochemical and pharmacokinetic parameters of DeltaTag (calculated parameters predicted by SwissADME, Supplementary Table T3; plasma stability and liver microsomal stability, Supplementary Fig. 6). With regards to the Lipinski rule of five, DeltaTag has a high molecular weight and its lipophilicity is at the borderline (Supplementary Table T3). While being stable

in mouse plasma, DeltaTag was almost completely metabolised within 50 min with an intrinsic clearance of 117 $\mu\text{L}/\text{min}/\text{mg}$ when tested in the presence of mouse liver microsomes (Supplementary Fig. 6). This intrinsic clearance of DeltaTag roughly translates to an *in vivo* clearance of $\sim 4 \text{ L}/\text{h}/\text{kg}$ using the simplified well-stirred model, which falls into the high clearance category (mouse liver blood flow 5.4 - 7.2 $\text{L}/\text{h}/\text{kg}$). We have discussed this limitation of metabolic instability in the manuscript – discussion section, page 24. The current pharmacokinetic profile of DeltaTag, while demonstrating potential for further optimisation, limits further *in vivo* validation of its biological efficacy. It is unlikely and also unethical for DeltaTag to be used in any animal experiments, given the high-costs and labour involved but without a clear potential for translational relevance.

Furthermore, although the authors have endeavored to clarify and correct specific NMR assignments and purity concerns, the fact that errors and ambiguities existed initially in the characterization data for multiple compounds inevitably raises persistent concerns regarding the overall rigor of the analytical validation.

We have carefully checked all data and all compounds are characterised according to the applicable standards.

Reviewer #2 (Remarks to the Author):

The authors have significantly revised the manuscript and provided thorough responses to all reviewer comments. Their revisions satisfactorily address all of my previous concerns, and the overall clarity and scientific rigor of the work have improved. I believe the manuscript is now suitable for publication without further changes.

We thank the Reviewer for the comment.

Reviewer #3 (Remarks to the Author):

The authors have addressed all my concerns, and I believe the manuscript has been improved to be published.

We thank the Reviewer for the comment.

Reviewer #4 (Remarks to the Author):

The manuscript entitled “Covalent Modification of a Glutamic Acid Inspired by HaloTag Technology” describes the rational design and characterization of covalent PDE δ inhibitors that selectively target a glutamic acid (E88) residue. Inspired by the HaloTag system, the authors developed DeltaTag, a prototype compound capable of covalently modifying PDE δ , thereby

disrupting the PDE δ –Rheb–mTORC1 signaling axis and impairing cancer cell proliferation. The study is comprehensive, integrating chemical synthesis, structural biology, proteomics, and cellular functional assays. Overall, the work represents a significant advance in expanding the scope of covalent inhibition strategies beyond cysteine/lysine residues, and it is of broad interest to the chemical biology and drug discovery communities. I believe it is suitable for publication in Nature Communications after major revisions.

We thank the Reviewer for the positive evaluation.

Comments:

1. While the authors briefly demonstrate preliminary covalent modification of UNC119, more extensive validation on additional targets would strengthen the claim that the approach is broadly applicable. Alternatively, providing additional structural/biochemical evidence on another protein target with a buried Glu/Asp residue.

We acknowledge the Reviewer's remark. We have adjusted texts in the manuscript to clarify that this study is proof-of-concept case study for a proposed methodology of covalent modification inspired by HaloTag technology at one glutamic acid of PDE δ , rather than generalising it to a broadly applicable technology. This is also in agreement with the corresponding remark of Reviewer 1 (comment 1).

Establishment of further examples would require the bioinformatic identification of further candidate proteins, their validation as drug targets and novel non-covalent inhibitor development including structural characterisation in the first place. Our previous work on PDE δ and UNC119 (PMID 23698361, PMID 27094677, PMID 28106325, PMID 27809361, PMID 27481943, PMID 28471079 and PMID 30956149) provides proof of this necessity, which laid the foundation for this paper now under review. The amount of work which has been documented in 7 publications clearly demonstrates that extensive validation on additional targets would be stand-alone investigations and would have to be reported in separate publications. In response to the Reviewer's initial request, we have provided orienting validation of the HaloTag design concept to covalently target UNC119. However, we consider further extensive validation on additional targets to be outside the scope of the current study.

In response to the Reviewer's suggestion and in acknowledgement of the Reviewer's argument, we have included additional structural analysis of potential protein targets with buried Asp/Glu residues suitable for covalent targeting (Supplementary Fig. S7). Hopefully this could provide initial insights for future investigations with the HaloTag inspiration for covalent carboxylate-targeted inhibition, with the aim of establishing the method's generalisability.

2. Although DNA alkylation and general cytotoxicity were tested, broader proteome-wide reactivity profiling (e.g., using chemoproteomics with activity-based probes) could further substantiate selectivity. At minimum, discuss potential risks of non-specific reactivity in more detail, especially given the electrophilic nature of alkyl bromides.

We thank the Reviewer for the suggestion. We have added the discussion for potential non-specific reactivity in much more detail in page 23 (also see below). DeltaTag has a much weaker reactivity towards the most reactive sulphur nucleophile glutathione ($t_{1/2}$ for GSH adduct formation = 1035 h), in comparison to reactive haloacetamides ($t_{1/2}$ for GSH adduct formation of chloroacetamides in the range of 10 min; $t_{1/2}$ for GSH adduct formation of chlorofluoroacetamides and dichloroacetamides in the range of 1000 min). Although off-target reactivity cannot be ruled out entirely, the current data suggest a potential of selective targeting of PDE δ at its binding site p.E88 by DeltaTag, with its structure adopted the nanomolar binding affinity and target specificity of Deltazinone 1 towards PDE δ .

“Structure-based design led to the discovery of DeltaTag with an alkyl bromide warhead, which covalently labels a glutamate residue in the prenyl binding pocket of PDE δ with sufficient reactivity. Given the electrophilic nature of alkyl bromides and potential risks of non-specific labelling at more reactive Cys and Lys side chains, we further investigated *in vitro* stability and labelling selectivity of DeltaTag. Gratifyingly, DeltaTag exhibits moderate stability, with low reactivity towards the most reactive sulphur nucleophile glutathione ($t_{1/2}$ for GSH adduct formation = 1035 h). *In vitro* labelling experiment with PDE δ further demonstrated that the labeling was selectively achieved at its binding site p.E88, as confirmed by mass spectrometric evaluation of the digested covalent adduct, with a parallel search for modifications at all nucleophilic residues including the more reactive Cys and Lys. In comparison to haloacetamides, particularly more recently developed chlorofluoroacetamides (CFAs) and dichloroacetamides (DCAs) with mild reactivity towards cysteines ($t_{1/2}$ for GSH adduct formation in the range of 1000 min) and consistently low, tuneable cytotoxicity,^{7,51,52} DeltaTag shows much weaker reactivity, suggesting at least comparably low toxicity from non-specific labelling reactivity. CFAs and DCAs also offer tuneable labelling selectivity and reversibility of covalency under aqueous conditions, which help eliminate or at least reduce off-target labelling of solvent-exposed residues.⁵² Similarly, alkyl bromide warhead targeting carboxylate residues may also hold promise for the development of targeted covalent ligands for its reduced off-target engagement with solvent-exposed residues, as the reversibility of ester bond formation could be significantly affected by protein microenvironment. In complex proteome under physiological conditions, sustained covalent modification of PDE δ was also achieved by DeltaTag despite the counteracting release of ligands from the binding site by Arl2. Phenotypically, DeltaTag inhibits proliferation of human cancer cell lines with an on-target activity supported by thermal proteome profiling and PDE δ knock-out cell model, and impacts the PDE δ -Rheb-mTOR pathway, at effective concentrations for which no off-target effects such as DNA alkylation or general cytotoxicity were observed.”

3. The manuscript demonstrates that DeltaTag overcomes limitations of reversible PDE δ inhibitors, but the comparison remains somewhat qualitative.

We acknowledge the Reviewer’s comment and agree that the comparison between reversible PDE δ inhibitors and DeltaTag remains somewhat qualitative. This proof-of-principle case study shows that the covalent PDE δ inhibitors based on the HaloTag substrates design principle present a qualitative improvement over the corresponding non-covalent inhibitors, as

demonstrated by the following observations. In qualitative terms the difference is clear; further structural optimisation, for example linker flexibility, might make the difference more substantial and increase quantitative measures but we do not want to speculate in this study.

The improvements of DeltaTag over reversible PDE δ inhibitors include:

- (1) higher thermal stability shifts of PDE δ upon binding as proven by NanoDSF – the difference is approximately 1.5-fold;
- (2) no inhibitor release by Arl2 upon covalent bond formation as demonstrated by fluorescence polarisation experiments (non-covalent inhibitors are readily released by Arl2) – the difference in IC₅₀ values recorded by FP assays is 2 to 3-fold;
- (3) sustained redistribution of Rheb over an extended time period, i.e. DeltaTag disrupts PDE δ -Rheb interaction and redistributes Rheb to endomembranes even overnight whereas redistribution of Rheb by reversible Deltazinone 1 is lost over this extended timeframe as demonstrated by immunofluorescence staining, see the new Figure 5c (page 19);
- (4) improved impact in cellular viability: e.g. no anti-proliferative activity up to 50 μ M demonstrated by the reversible Deltazinone 1 in KRAS mutant and dependent MIA PaCa-2 cells as compared to IC₅₀ for DeltaTag of 7 μ M – the difference is > 7-fold. Improvement in other cell lines however remain qualitative.

Of particular note to the qualitative improvement in cellular potency of covalent DeltaTag over reversible inhibitors, we and others (the Abankwa group, the Ismail group and the Sheng group) have reported and consistently observed PDE δ inhibitors with nanomolar affinity *in vitro* but only micromolar IC₅₀ *in cellulo*. Even PDE δ degraders (PMID 31829492, PMID 38229750, PMID 38229750, PMID 39146536, PMID 35127385) that can act sub-stoichiometrically to PDE δ showed with comparable micromolar IC₅₀ *in cellulo*, similar to the cellular potency of DeltaTag reported in this study. We do not want to speculate any reasoning for this observation but further structural optimisation of PDE δ inhibitors could potentially remain as qualitative improvements, with cellular IC₅₀ values in the low nanomolar range.

4. The covalent modification is demonstrated at E88, but it remains unclear why this residue reacts preferentially over others. Consider including computational modelling or additional mutagenesis data (e.g., E88Q, E89Q) to further validate selectivity.

The preferential reactivity of the compound **5e** and **6a** towards p.E88 instead of p.E89 is clear because p.E88 has its side chain pointing into the binding pocket of PDE δ while p.E89 has its side chain surface exposed. The orientations of p.E88 and p.E89 are opposite each other and it only makes sense that p.E88 is targeted preferentially over p.E89 with the compounds occupy the prenyl-binding site of PDE δ . The selective reactivity towards p.E88 over other nucleophilic residues is due to spatial proximity, as a general basis for rational TCI development. The

glutamate residue p.E88 is currently the only reported nucleophilic residue of PDE δ accessible for covalent targeting. We have previously performed labelling experiments with PDE δ E88A mutant protein with a covalent inhibitor assuming similar positioning and orientation in PDE δ (PMID 28434875) as compared to **5e** and **6a** in this study. Notably, PDE δ E88A mutant protein was not labelled with the covalent inhibitor, even when presented in high excess. With optimised orientation of the compound **6a** towards p.E88 of PDE δ , there is no reactivity observed for other nucleophilic residues in the vicinity of the covalent warhead, hence demonstrating *in vitro* labelling selectivity. We apologise that we did not point this out clear enough and we thank the Reviewer for the suggestions. We have now included additional texts in page 10-11 and visual representation of the orientations of p.E88 and p.E89 in Figure 3c and 3d.

5. The absence of *in vivo* evaluation is a limitation. However, the authors could strengthen the translational relevance by Reporting preliminary pharmacokinetic stability (microsomal stability, plasma stability, or logD) of DeltaTag.

We thank the Reviewer for the suggestions. We would like to first clarify that the best compound presented in this study was not developed with the intention of advancing PDE δ inhibitors for drug discovery. It is rather a proof-of-concept case study, demonstrating a carboxylate-targeting strategy inspired by the design principle of HaloTag substrates (and also confirmed by the Editor, that *in vivo* validation of antitumor activity of covalent inhibitors are not required). We do not intend to claim any translational potential of the findings presented in this study for anti-cancer activity.

We have, however, included preliminary assessment of physiochemical and pharmacokinetic parameters of DeltaTag in Supplementary Table T3 (calculated parameters predicted by SwissADME) and Supplementary Fig. S6 (plasma stability and liver microsomal stability). With regards to the Lipinski rule of five, DeltaTag has a high molecular weight and its lipophilicity is at the borderline (Supplementary Table T3). While being stable in mouse plasma, DeltaTag was almost completely metabolised within 50 min with an intrinsic clearance of 117 $\mu\text{L}/\text{min}/\text{mg}$ when tested in the presence of mouse liver microsomes (Supplementary Fig. 6). This intrinsic clearance of DeltaTag roughly translates to an *in vivo* clearance of $\sim 4 \text{ L}/\text{h}/\text{kg}$ using the simplified well-stirred model, which falls into the high clearance category (mouse liver blood flow 5.4 - 7.2 $\text{L}/\text{h}/\text{kg}$). We have discussed this limitation of metabolic instability in the manuscript – discussion section, page 24. The current pharmacokinetic profile of DeltaTag, while demonstrating potential for further optimisation, limits further *in vivo* validation of its biological efficacy. It is unlikely and also unethical for DeltaTag to be used in any animal experiments, given the high-costs and labour involved but without a clear potential for translational relevance.

Problems of Structural Determination

a) Several compounds (3a, 3c, 3d, 4a, 4b, 4c, 4d, 5a, 5b, 5c, 5d, 5e, 5f, 5g, 5h, 5i, 5j, 5k, 5l, 6a, 6b) exhibit multiple unexplained signals at δ 8.0–9.0 in their ^1H NMR spectra, suggesting the presence of consistent impurities and indicating inadequate compound purity.

^1H NMR Spectrum of 3a (600 MHz, $\text{DMSO}-d_6$)

^1H NMR Spectrum of 4a (600 MHz, $\text{DMSO}-d_6$)

^1H NMR Spectrum of 5a (700 MHz, $\text{DMSO}-d_6$)

^1H NMR Spectrum of 6a (700 MHz, $\text{DMSO}-d_6$)

b) **Page 39.** The ^1H NMR spectrum of compound **2a** does not match its proposed structure (missing a CH_2 group)

^1H NMR Spectrum of **2a** (600 MHz, $\text{DMSO}-d_6$)

c) **Page 94.** The ^{13}C spectrum of compound **6b** contains unassigned peaks at δ 176, 77, and 67, indicating potential impurities or low purity.

¹³C NMR Spectrum of **6b** (176 MHz, DMSO-*d*₆)

- d) ¹H NMR spectra for compounds 2a, 2b, 2c, and 2d lack signals corresponding to -NH groups.
- e) The ¹H NMR assignment for compound S5 is incorrect, with the carboxyl (-COOH) peak missing from the reported data.
- f) The ¹H NMR assignment for compound S4 contains errors: 2.52 (s, 11H).